# Development of Trade and Financial-Economical Relationships between China and Russia: A Study Based on the Trade Gravity Model

Gleb Aksenov [1], Ronglin Li [1], Qamar Abbas [2,*], Houlda Fambo [1], Sergey Popkov [3], Vadim Ponkratov [4], Mikhail Kosov [5,6,7], Izabella Elyakova [8] and Marina Vasiljeva [9,10]

1. Institute of International Economics, Nankai University, Tianjin 300071, China
2. College of Finance and Economics, Jiangsu University, Zhenjiang 212013, China
3. Department of Financial Management and Financial Law, Moscow Metropolitan Governance Yury Luzhkov University, 107045 Moscow, Russia
4. Center of Financial Policy, Financial University under the Government of the Russian Federation, 125167 Moscow, Russia
5. Department of State and Municipal Finance, Plekhanov Russian University of Economics, 115093 Moscow, Russia
6. Legal Management Institute HSLA, HSE University, 101000 Moscow, Russia
7. Department of Public Finances, Financial University under the Government of the Russian Federation, 125167 Moscow, Russia
8. Department of Economics and Finance, Financial and Economic Institute, M.K. Ammosov North-Eastern Federal University, 677027 Yakutsk, Russia
9. Autonomous Non-Profit Organization "Publishing House Scientific Review" (Nauchnoe Obozrenie), 127051 Moscow, Russia
10. Atlantic Science and Technology Academic Press, Boston, MA 01233, USA
* Correspondence: q.abbas47@outlook.com

**Abstract:** This article empirically assessed new opportunities and provides a conceptual justification for promising areas of trade and financial and economic relations between China and Russia amidst ongoing global turbulence, the post-COVID situation, and sanctions pressure. The study utilized the trade gravity model, taking into account the latest trends in the development of the research subject and object, as well as current challenges and trends in the global economy. The study revealed similarities between the political systems, reforms, and policies of China and Russia, with centralized power structures overlapping and supporting each other at international forums such as the UNSC. The findings suggest that both countries plan to increase trade volume in the next two years, with China focusing more on trade and economic development, while Russia works to promote security and political stability. This study provides valuable insights into the economic relationship between Russia and China, its impact on the US and Europe, and highlights the need for effective policy interventions to address the challenges posed by this relationship. It offers significant theoretical and practical contributions, including the potential to unlock the potential of national economies, increase their competitiveness and help states enter a phase of advanced and sustainable development. This article provides several policy recommendations to ensure the long-term sustainability of the economic relationship between Russia and China and foster mutual understanding and trust between their peoples. These include promoting trade diversification, enhancing financial cooperation, addressing trade barriers, strengthening political and security coordination, mitigating negative impacts on other countries, promoting sustainable development, and fostering people-to-people exchanges.

**Keywords:** gravity model of international trade; sustainable development; financial-economical relationship; COVID pandemic-19; international sanctions; trade policy; China; Russia; dependence on imports of Chinese goods; asymmetrical interdependence

# 1. Introduction

As certain countries continue their attempts to isolate Russia on the world stage, Russia seeks to strengthen economic, military, and political ties with China, Russia's strategic partner [1,2]. In 2022, the two countries intensified and increased their economic cooperation [3,4]. China, as the world's leading economy, has long been Russia's largest trading partner [5]. The complementarity of the two countries' economies, a common border, large-scale transportation projects, cooperation in key industries, participation in BRICS and the SCO, and a number of other factors all serve as major incentives for current and new bilateral initiatives [6]. The possibility of unlocking the potential of these relations is largely determined by their current content and current trends [7].

Russian fertilizers, timber and food products are no less valuable to China than raw materials. China's rapid economic and social development over the past few decades has led to an exponential increase in the demand for raw materials and food products, which China buys on foreign markets. Russia has become China's largest supplier in such products as frozen fish (29.6% of Chinese imports in 2021), fertilizers (28%), wood products (16.7%) and fuel (coal, oil, petroleum products and gas combined; 13%). Russia also plays a leading role in China's import market for asbestos, electricity, magnesium hydroxide, buckwheat, newsprint, linseed, sunflower oil, etc. In 2022, China's demand for Russian goods, including non-raw goods such as semi-finished unalloyed steel products, synthetic rubber, frozen fish, and rapeseed oil, increased significantly [8].

The diversification of exports and increase in added value in Russian exports to China remain top priorities today. The importance of the further development of services exports, particularly in the field of information technology, finance, tourism, and telemedicine, is growing rapidly.

Regional optimization of logistics plays a key role in the expansion of exports; in particular, existing routes for the delivery of goods from the central regions of Russia and Western Siberia to the provinces of Central and Western China are being improved. Infrastructure for new rail routes through Kazakhstan is currently under construction. The attractive idea of turning the Northern Sea Route into part of the Polar Silk Road and the need to integrate it with the existing Eurasian land routes offer ample opportunities for cooperation between Russia and China in the coming decades.

In the current geopolitical environment, deepening strategic cooperation with China is in Russia's interest. In 2023, Russia will continue to develop its partnership with China. But how will other world players respond to Russia-China growing comprehensive cooperation?

In 2023, Russia has increased its energy exports to China. In November 2022, the Russian Federation overtook Saudi Arabia as the main supplier of crude oil to China and became one of the leading gas exporters to China. The Power of Siberia-2 pipeline, which could provide China with nearly 50 billion cubic meters of gas per year, is currently in the contract process [9].

Yet even the existing energy infrastructure connecting the two neighboring countries has allowed Russia to increase its natural and liquefied gas supplies to China. Meanwhile, Chinese coal imports from Russia have reached a five-year high, and daily trading volumes for the yuan-ruble exchange rate have increased significantly in 2022. Thus, bilateral ties between Russia and China, especially in the economic and energy sectors, are likely to expand in 2023.

Considering that bilateral trade between Russia and China has grown by 25% by the end of 2022, "we will be able to reach the $200 billion target mark set by us for 2024 ahead of schedule", Putin said during talks with Chinese President Xi Jinpin [10].

Europe's energy crisis remains difficult to ease in 2023. The energy crisis and Europe's high inflation would surely worsen its economic prospects [11]. According to the European Commission, economic activity in the EU, the eurozone, and most member states are expected to continue to contract in the first quarter of 2023 "We expect one-third of the world economy to be in recession" [12].

Due to the above, Russia must enter new strategic consumer markets, strengthen its position in production chains, transport corridors, and international financial organizations. Russia needs a strategic partnership with China to form a "Greater Eurasia". Simultaneously, Russia must learn the right lessons from the unsuccessful experience of creating a Greater Europe to avoid repeating the same mistakes in its relations with economically powerful China [13]. The emerging strategic partnership between the two countries requires Russia to create a viable counterweight to China. Otherwise, the emerging asymmetric interdependence will create a temptation for China to demand political concessions [14]. In the long term, it will make this partnership unprofitable for Russia. The urgent tasks for Russia at this stage are the "soft" balancing of China, which involves creating a balancing act and preventing the zero-sum games that put an end to the Greater Europe project [15].

Thus, in modern conditions, it is important to study the prospects of sustainable development of trade and financial-economic relations between China and Russia and to develop proposals on forming vectors of smoothing asymmetrical interdependence in double-sided relationships under conditions of international sanctions [16]. This research contributes to the literature by examining the Sino-Russian trade relationship and its impact on the US and Europe under international sanctions. The escalation of sanctions and geopolitical tensions between Russia and some countries of the world has led to an increased focus on China's role in the Eurasian region. The current study is motivated by the need to assess empirically the new opportunities and provide a conceptual justification for promising areas of trade and financial and economic relations between China and Russia. While there is a growing body of literature on Sino-Russian relations, most studies have focused on the political aspects of the relationship, with limited attention paid to the economic and trade dimensions. There is a need for empirical studies that examine the trade relationship between China and Russia, its impact on the US and Europe under international sanctions, and the potential implications for the global economy. Therefore, the current research will answer the following research questions:

- What are the new opportunities and promising areas of trade and financial and economic relations between China and Russia under ongoing global turbulence, the post-COVID situation, and sanctions pressure?
- How can these opportunities be assessed empirically?

This research offers significant theoretical and practical contributions to the literature on Sino-Russian relations. By empirically assessing new opportunities and providing a conceptual justification for promising areas of trade and financial and economic relations between China and Russia, the study offers valuable insights into the potential of national economies, increasing their competitiveness and helping actively interacting states to enter a phase of advanced and sustainable development. The study's findings on the impact of the Sino-Russian trade relationship on the US and Europe under international sanctions will provide policymakers with valuable information for developing effective policies to address the challenges posed by this relationship. Overall, this research fills a gap in the literature by providing a comprehensive analysis of the Sino-Russian trade relationship and its impact on the global economy, thus contributing to a better understanding of the complex interplay between economic and political factors in international relations.

## 2. Theoretical Framework

The economic relationship between Russia and China has been a topic of growing interest recently. Scholars have examined various aspects of this relationship, including its drivers, implications, and challenges, as well as its impact on global politics and economics.

One trend in the literature is the focus on the drivers of the Sino-Russian economic relationship. According to Zhou and Liu [17], the main drivers of this relationship include complementary economic structures, geopolitical considerations, and mutual benefits. Another study by Zhang and Sun [18] argues that the drivers of this relationship are also shaped by China's Belt and Road Initiative (BRI) and Russia's Eurasian Economic Union (EEU), which provide a framework for their economic cooperation.

Another trend in the literature is the examination of the implications of the Sino-Russian economic relationship. For example, Lai and Zhao [19] explore the impact of this relationship on the US-led international order, arguing that it challenges the existing global power structure and poses a threat to US hegemony. Similarly, Kirillova and Lourié [20] examine the impact of this relationship on the European Union (EU), arguing that it could lead to a reorientation of the global economic order and a shift in power from the EU to China and Russia.

Furthermore, scholars have also examined the challenges and opportunities of the Sino-Russian economic relationship. A study by Fan and Wu [21] highlights the challenges of the Sino-Russian economic relationship, including the lack of trust, cultural differences, and trade barriers. In contrast, a study by Zhang and Sun [18] emphasizes the opportunities of this relationship, including the potential for infrastructure development, energy cooperation, and innovation collaboration.

Overall, the literature on the Sino-Russian economic relationship is multifaceted, covering various aspects of this relationship, including its drivers, implications, challenges, and opportunities. This study contributes to this literature by examining the sustainable development of this relationship and providing policy recommendations for its long-term sustainability.

### 2.1. The Sino-Russian Trade and Financial-Economic Relationship

Simola [22] investigated the economic dependency of China and Russia regarding their economic position and found that both countries mostly rely on traditional trade considering comparative advantages [23]. Russia is rich in terms of natural resources such as oil and gas, which is the reason for China's dependency increase on Russia in terms of importing its oil and gas [24]. Additionally, Russia has a competitive advantage over China with regard to the energy sectors; the energy industry is considered the backbone of any economy where the decisions are made based on the consumer perspective [25,26]. Furthermore, the findings of Meynkhard [27] indicate that the Russian price of oil and gas to China is based on the market. China's exports of gas and oil from Russia have been increasing gradually.

China's spending on Russian commodities has also increased in the last couple of years. Also, Russia has more desire and importance for an economic relationship with China than China's preference [28]. In addition, the overall economic trade volume between China and Russia in 2021 was around $147 billion, which is expected to grow further in 2022 [29]. Besides, the economic relationship of both countries can be analyzed with the long-term economic approach as they are concerned with their future, which is the reason they initiated the China-Mongolia-Russia economic corridor, which can improve their economic relationship further as it can facilitate improving the connectivity and trade relationships between both of these countries [30]. To further delve into this, Megits [31] conducted research to evaluate and investigate the impact of the Russia-China trade and economic affiliation on the U.S economy [32]. The economic ties of Russia and China have the capacity to destruct the market dominance of the U.S.A [33]. Therefore, the United States should closely monitor and observe their relationship [34].

The historical findings of the sustainable economic relationship between Russia and China by Chen and Bao [35] expressed that from the beginning of the 21st century, both countries tried expanding their bilateral relationship, which has a significant impact on the sustainable economic growth of both regions. Hence, the findings of Mahlstein et al. [36] also indicate that sustainable economic development also has a significant impact on the overall economy of the world. Table 1 summarizes main research directions on China-Russia trade in existing literature.

**Table 1.** Main research directions on China–Russia trade.

| Research Direction | Author, Year | Outcome |
|---|---|---|
| 1. China-Russia dual-side trade | Nan, G (2022) [14] | Russia and China have a potential for extending border trade |
| | | Russia and China complete each other in international trade |
| 2. China-Russia investment cooperation | Ann F. Ostrovsky (2000) [37] | Russia must improve the investment climate in the Far East and Siberia |
| | Li Shizhen (2018) [38] | China and Russia must provide more opportunities for developing private business in both countries |
| 3. Trade policy of China and Russia | Michael A. Hitt and David Ahlstrom (2004) [39] | China and Russia must cooperate to achieve fast economic development |
| 4. National security of China and Russia | Marcel de Haas (2019) [40] | Aside from reinforcing economic and trade cooperation, it is necessary to reinforce national security cooperation as well |
| | Titarenko (2005) [41] | Rejects "Chinese threat theory" |

*2.2. The Impact of International Sanctions on Trade Relations between Russia and China*

Due to sanctions and unfavorable economic conditions, Russian foreign trade has undergone significant changes since 2014. The share of EU countries in Russia's foreign trade reduced to minimum, while the share of Asia-Pacific countries, especially China, increased significantly [42]. Anti-Russian sanctions have not only affected the commodity markets but also resulted in tighter conditions in global financial markets, reflecting greater risk aversion and uncertainty. The IMF's April 2022 forecast shows a slowdown in global GDP growth for the eurozone and the United Kingdom in 2023 (see Table 2).

**Table 2.** The impact of the anti-Russian sanctions and Ukrainian conflict on the GDP dynamics in Western countries (according to PPP as a percentage over the previous year) [43,44].

| | Assessed as of April 2022 | | | Assessed as of January 2022 (for Reference) | | |
|---|---|---|---|---|---|---|
| Countries | 2021 | 2022 | 2023 | 2021 | 2022 | 2023 |
| The world as a whole, incl. | 6.1 | 3.6 | 3.6 | 5.9 | 4.4 | 3.8 |
| Developed countries, incl. | 5.2 | 3.3 | 2.4 | 5.0 | 3.9 | 2.6 |
| USA | 5.7 | 3.7 | 2.3 | 5.6 | 4.0 | 2.6 |
| Euro area | 5.3 | 2.8 | 2.3 | 5.2 | 3.9 | 2.5 |
| UK | 7.4 | 3.7 | 1.2 | 7.2 | 4.7 | 2.3 |
| Japan | 1.6 | 2.4 | 2.3 | 1.6 | 3.3 | 1.8 |
| Canada | 4.6 | 3.9 | 2.8 | 4.7 | 4.1 | 2.8 |

The effectiveness of applying economic and other types of sanctions is a debatable issue in the scientific literature. Some researchers believe that economic coercion measures can cause appreciable damage to the country, which is subject to restrictions, and force its leadership to change its policy [45–47]. Others draw attention to the ineffectiveness of economic sanctions as a foreign policy tool [48,49]. In particular, Hufbauer [48] and Morgan [50], who evaluated the effectiveness of sanctions over a long period, concluded that sanctions are effective only in 1/3 of these cases. Many researchers have used the gravity model to study the effects of sanctions [51–54].

Sanctions have complicated the development of the Russian economy and adversely affected trade with traditional partners, mainly the EU [55–60]. However, some researchers argue that foreign economic relations are not actually shifting to the East and that trade flows will revert to traditional partners in the case of the termination of sanctions or the effect of ruble devaluation [61,62]. Financial restrictions have had the most significant nega-

tive impact of all sanctions on the Russian economy, as highlighted in various publications by Russian economists. [63–65].

Mahlstein et al. [36] found that Russia's real Gross Domestic Product experienced a loss of over 14% due to the sanctions, resulting in increased trade costs and raw material unavailability. While non-associated economies benefited, China's participation in the embargo caused a significant economic loss for Russia. Firms' performance in Russia declined, as the demand for Russian commodities decreased, and trading patterns looked for alternatives [66]. IMF and World Bank reports suggest a significant decline in imports and exports for Russia in 2022, with a potential increase in imports in 2023, but with exports continuing to drop. Figure 1 provides a graphical illustration of these trends.

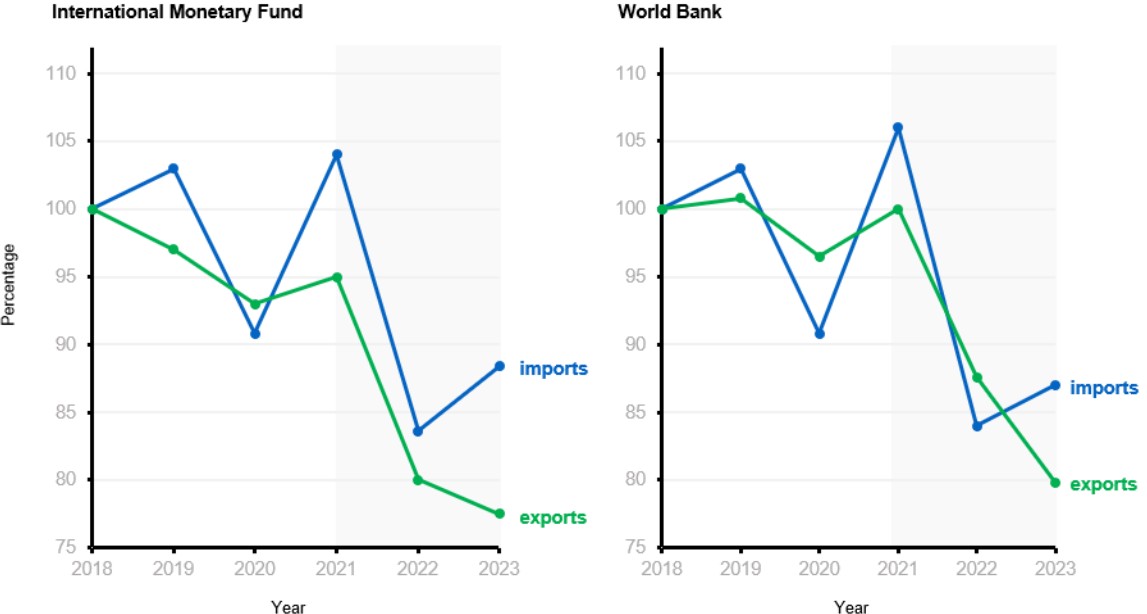

**Figure 1.** Trends of Russian Imports and Exports [67,68].

Huynh [69] found that international sanctions negatively impacted the non-energy sectors and firms, leading to a decline in their performance and growth. These sanctions also had negative economic effects on Russia's research and development and capital expenditure. Liadze et al. [70] demonstrated that consistent international sanctions against Russia could reduce global GDP by 1% by the end of 2023, and increase inflation in Russia by 20% [71–73]. The predicted Gross Domestic Product of the UK can decline by 0.5 percent in 2023 due to these sanctions [74]. While Kholodilin and Netšunajev [75] found no strong evidence to suggest that the sanctions would cause a decline in the GDP of Russia or associated countries. The EU Council, the World Bank, and IMF forecasted a drop in Russian GDP for 2022, with the IMF suggesting the largest decline of 4.5% (Figure 2).

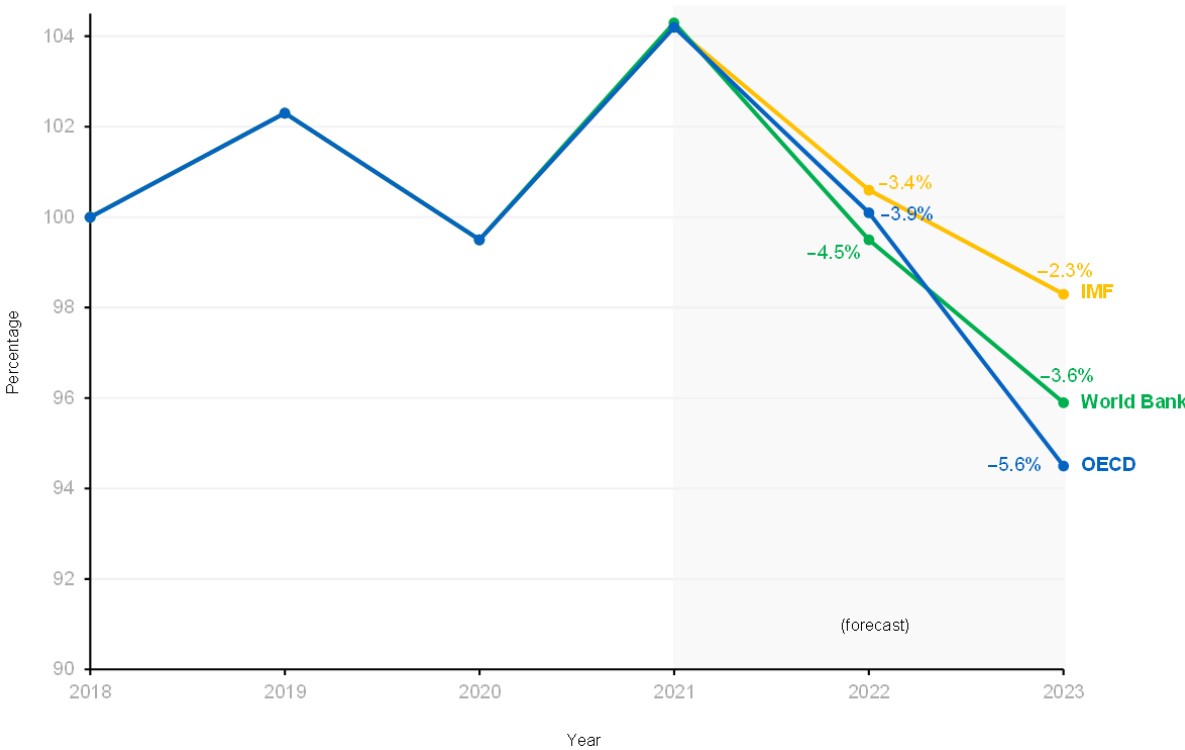

**Figure 2.** GDP of Russia [68].

The recent sanctions on Russia had negative impacts on both the Russian economy and the world, with increased energy prices and risks to the global economy [76]. European countries faced a rise in the consumer price index, specifically in energy prices, affecting many industries and resulting in higher production costs for consumers. This inflation also affected Russia, as shown in Figure 3, where the IMF, OECD, and World Bank predicted a spike in the inflation rate, reaching up to 14%, with fluctuations continuing into 2023.

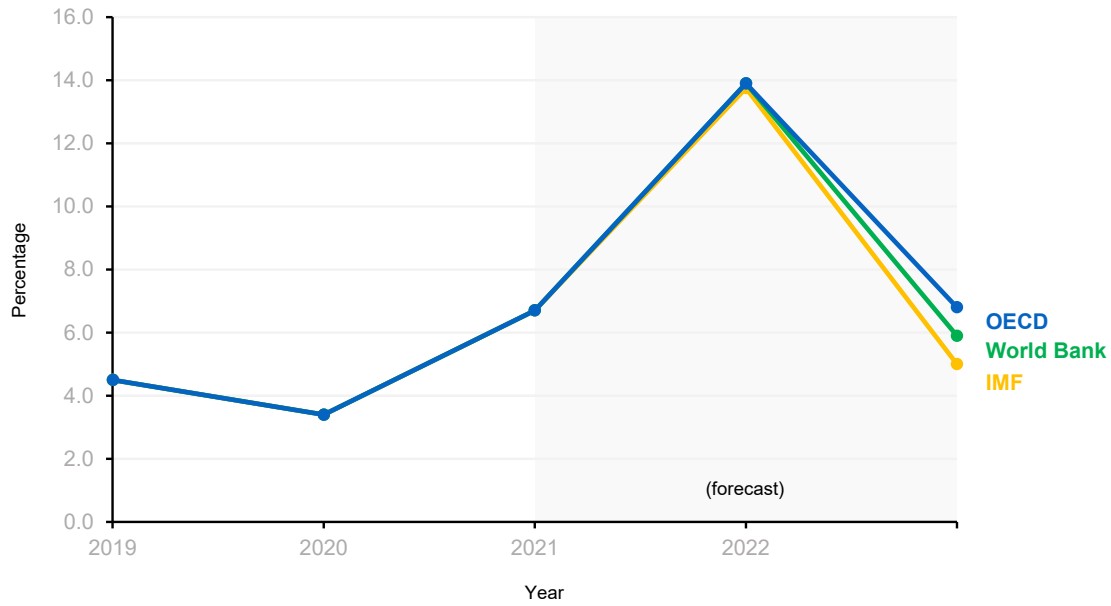

**Figure 3.** Inflation Rate in Russia [67,68,77].

Sanctions in Russia have impacted imports of foodstuff, reducing them by almost 50% in the first two years. This will likely disrupt the agricultural sector, particularly in seeds, pesticides, agricultural technologies, and veterinary medicines, as Russia is highly dependent on imports. The news of sanctions also decreases the future returns of agricultural commodities. These findings indicate that the sanctions will decrease economic performance and require restoring significant financial resources.

### 2.3. The Asymmetric Interdependence as the Basis for the Development of Financial and Economic Cooperation between Russia and China

The Theory of Complex Interdependence has got its development in foreign affairs. The term "complex interdependence" was proposed by Raymond Leslie Buell in 1925 to describe a new order of relations between countries, economies, and different cultures [78]. The theory of complex interdependence was formulated by R. Keohane and J. Nye in 1987 [79]. Complex interdependence occurs because of expanding financial ties and world trade between countries, leading to intertwined issues in international relations [79]. The importance of military force and coercive mechanisms decreases as countries find common interests in problem-solving. Increased economic and other forms of interdependence enhance the possibility of positive partnerships and interactions between nations.

Asymmetric interdependence is a factor that exacerbates financial, political, and other risks and increases uncertainty in matters of financial and economic cooperation between countries [80,81]. On the one hand, the complementarity of the basic sectors of Russia and China's economies objectively contributes to an increase in cooperation and trade. On the other hand, the asymmetric interdependence of Russia and its foreign trade partners, in particular China, contributes to the conservation of an archaic model of mutual trade based on Russian raw material exports and Chinese imports of manufacturing products [14].

Trading ties between China and Russia have strengthened even more over 2020–2021 (see Figure 4), but bilateral trade dependence is disproportionate, as shipments to China account for about 18% of Russian exports, but less than 3% of China's total imports. Additionally, imports from China account for nearly 24% of Russian imports, but only about 2% of China's total exports [82].

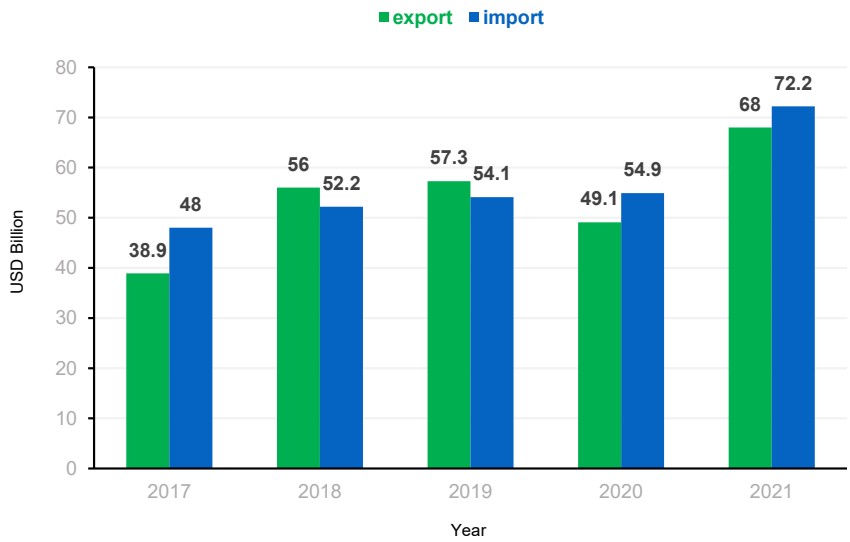

**Figure 4.** Dynamics of Sino-Russian trade turnover in 2017–2021 [82].

The Russian-Chinese trade turnover in January-October 2022 increased by 33% compared to the same period of the previous year, up to a record $153.9 billion. More than 50% of China's exports to Russia now fall into the categories of equipment, mechanical devices, electrical machines, electronic equipment, and land transport (groups 84–87 of the international commodity nomenclature for foreign economic activity) [82].

Strengthening of multilateral sanctions against Russia has raised issues about the level of trade integration between Russia and China in the market, focusing on whether Russia can divert its exports there. Russian-Ukrainian geopolitical tensions and the resulting sanctions have put China in an awkward position of having to manage its trade and financial ties with Russia while reducing the risk of worsening its relations with the US and the EU [83].

China is considering buying or increasing its stakes in Russian energy and natural resources companies such as Gazprom and Rusal [84], but negotiations are still at an early stage. While Russia is China's most significant supplier of energy resources, China has already become the main supplier of technologies for industry and Russia is more dependent on China for high-value commodities, including engineering products, electronics, and consumer goods [85].

On 24 February 2022, the US announced new sanctions against Russia [86], which, among other things, are aimed at limiting technology exports. The sanctions apply to the export of semiconductors, computers, telecommunications equipment, lasers, sensors, and aircraft components to Russia. Russian companies had to look for a replacement, and China could partially provide it. However, US restrictions apply not only to technology imported directly from Western countries, but also to goods manufactured in any country, if they use American intellectual property. This applies in particular to chip manufacturers, such as Taiwanese TSMC and Shanghai-based chip maker SMIC. The Taiwanese TSMC has already refused to supply semiconductors to Russian companies [87]. If SMIC and other mainland Chinese companies continue to export their products to Russia, they could be cut off from US technology [88]. Due to similar sanctions, Huawei suffered in its time. However, SMIC is already under US sanctions in part.

Russia's trade dependence on China has been increasing, particularly in supplying raw materials, technologies, and components for cars and semiconductors. The Russian financial system has been investing in the yuan even before sanctions were imposed. The Central Bank and the National Welfare Fund currently hold about $140 billion in Chinese bonds, almost a quarter of foreign ownership in China's domestic bond market [89]. The Chinese currency accounted for 13.1% of the Russian Central Bank's foreign exchange reserves in June 2021, compared to 0.1% in June 2017, with Moscow's dollar holdings dropping to 16.4% from 46.3% in the same period [90].

China and Russia are trying to decrease their financial system's reliance on the West. They began using their currencies for bilateral trade in 2010. Yuan payments made up 28% of Chinese exports to Russia in H1 2021, up from just 2% in 2013. In 2014, a 150-billion-yuan swap agreement was signed and extended by 150 billion yuan in 2020 [91]. However, there are constraints on the usage of Russian and Chinese currencies in trade because of the ruble's geopolitical volatility and the yuan's lack of full convertibility.

Russia can use its yuan holdings and its own cross-border payment systems to counter the impact of Western sanctions. However, in the sphere of finance, Russia is hardly a priority market for China. The Asian Infrastructure Investment Bank, whose main shareholder is China, has already stopped cooperation with Russia [92].

The following outcomes can be made from the literature analysis: (1) there are perspectives for developing China-Russia trade, including the border one; (2) there still is major potential for China-Russia investment cooperation; (3) the trade policy of Russia and China in 2022 has reinforced their cooperation in various areas; (4) national security is a basis for economic trade between the two countries, while economic cooperation and national security interact with each other; (5) the difference between the advantages of manufacturing industry in China and Russia is reflected in the difference in labor and technological intensity levels.

Therefore, in the process of international financial and economic cooperation, it is important to consider and apply the necessary measures to minimize various risks arising under the influence of the asymmetric interdependence of countries. In modern conditions, the dialectic of asymmetric interdependence–from the imbalance of dependencies to its

restoration–is an objective basis for developing a mutually beneficial economic partnership between countries, including in relations between Russia and China, both on a bilateral basis and in a multilateral format.

## 3. Methodology and Data

### 3.1. Model Specification

The gravity model of international trade in international economics is a model that, in its traditional form, predicts bilateral trade flows based on the economic sizes and distance between two units. Research shows "overwhelming evidence that trade tends to fall with distance" [93,94].

The gravity model possesses considerable robustness and explanatory power; these features enabled numerous researchers to study the trade flow effects of a wide variety of real or dummy explanatory variables (Table 3), including country-specific characteristics (GDP, population, and income) and bilateral characteristics (the geographical distance between exporter and importer).It was useful to investigate variables incorporating the drivers of and barriers to trade, including geographical contiguity, ethnic ties, linguistic identity, colonial links, island or landlocked status, exchange rates, tariff and non-tariff barriers, currency unions, trade agreements, and common trade unions [95–101]. Furthermore, the gravity model is used to measure trade efficiency or trade potential by calculating the differences between predicted and observed trade flows [102–108].

**Table 3.** Description, expected signs, and theoretical analysis of explanatory variables. Developed by authors based on certain data from [14].

| Explanatory Variables | Variable Description | Expected Sign | Comment |
|---|---|---|---|
| $T_{ijt}$ | Total bilateral trade between China $i$ and trading partner $j$ in year $t$ | + | If the total volume of bilateral trade continues to increase, it means that trade between the two countries is developing in a positive trend. |
| $Y_{it}Y_{jt}$ | Product of China's GDP $i$ and trading partner $j$ in year $t$ | + | GDP reflects the total economic volume and scale of a country or region. The larger the economic scale, the greater the supply and demand potential, and the greater the volume of bilateral trade. |
| $I_{ijt}$ | Direct investment of trading partner country $j$ in China $i$ in year $t$. | + | FDI promotes bilateral trade: the more investment, the more bilateral trade. |
| $GDP_{ijt}$ | The absolute value of the difference in GDP per capita between China $I$ and trading partner $j$ in year $t$ | - | The greater the difference, the greater the inter-industry trade between the two countries, and the lower the volume of bilateral trade. |
| $D_{ij}$ | Geographical distance between China $i$ and trading partner $j$ | - | The longer the distance, the higher the cost of transportation, and the lower the volume of bilateral trade. |
| $Ȿ_{ijt}$ | Signing of a free trade agreement by China $I$ with trading partner country $j$ in year $t$ | + | The signing of free trade agreements and the reduction in trade barriers will help increase bilateral trade between the two countries. |
| $B_j$ | Indicates whether the *One Belt One Road* initiative advances | + | The *One Belt One Road* policy proposal promotes mutual benefits between the two countries, thereby increasing the volume of bilateral trade between the two countries |

**Table 3.** *Cont.*

| Explanatory Variables | Variable Description | Expected Sign | Comment |
|---|---|---|---|
| Cov*jt* | The impact of the pandemic on the total volume of bilateral trade between China *i* and Russia *j* in year *t* (2020–2022) | + | Sino-Russian trade cooperation has been affected by COVID-19: in June 2020, the trade volume fell by 5.6% and amounted to 46.19 billion US dollars. Exports to Russia in the first half of 2020 fell by 6% and amounted to 20.94 billion US dollars. Along with this, deliveries from Moscow to Beijing decreased by 5.3% (to USD 28.22 billion) [103]. Generally, at the end of 2020, the trade between Russia and China decreased by 2.9% in annual terms. However, the period 2020–2022 can be regarded as a minor impact of the epidemic on trade and economic relations. |
| S*jt* | China reported a decrease in exports of goods to Russia in 2022 by 26% (from USD5.126 billion to USD3.8 billion). This is caused by both sanction restrictions and related problems with logistics | + | The variables are exporter (*i*), partner country (*j*), type of product (*k*), and time (*t*). Russia is the exporter in all observations, and the type of goods may be sanctioned. The dependent variable is the volume of trade in sanctioned or non-sanctioned goods from Russia with its trading partner *j* in month *t*, measured in millions of dollars. Sanctions is a dummy variable that takes a value of 1 if the export flows consist of prohibited product groups; otherwise, it equals 0, acting as a criterion for classifying goods in a group. The total volume of trade in sanctioned goods is subtracted from non-sanctioned goods to create an appropriate control group and use an estimate of the difference in the sample. As a result, it is possible to determine how the dynamics change in the context of groups and whether there are differences between them. |

"+"—expected positive impact of variable on bilateral trade between China and Russia; "-"—expected negative impact of variable on bilateral trade between China and Russia.

To analyze the factors affecting the bilateral trade flow, the gravity model of international trade generally applies. The model originates from Newton's law of universal gravitation in physics. As economics has grown, more and more scientists started to apply the gravity model to economics. Tinbergen [94] and Pöyhönen [109] were the first to use the so-called gravity equation to study international trade. Both scholars used the gravity model in 1962 and 1963, respectively, to apply economic aggregates and geographic distances of the two countries to study issues affecting bilateral trade flows. Subsequently, as the research deepened, more and more explanatory variables were introduced into the model to meet the needs of the study, for example, variables related to economic factors, including GDP per capita, national income per capita, foreign direct investment, tariff rates, consumer price index, etc.; those related to geographical factors, including climate, territorial borders, etc.; variables related to demographic factors, including language, religion, population size, etc.; political factors, including the level of the country's development, whether to sign a free trade agreement, whether an FTA exists, etc. Because of its simple principle and availability of reliable data, the trade gravity model gradually became widely used in economic research and became one of the main empirical tools for studying trade flows in international trade.

The basic form of the trade gravity model:

$$T_{ij} = A \frac{(Y_i^\alpha Y_j^\beta)}{D_{ij}^\gamma} \tag{1}$$

$T$ is the current volume of bilateral trade between the country *i* and the country *j*,

$A$ is a constant term,

$Y_i^\alpha$ is the GDP of country *i*, $Y_j^\beta$ is the GDP of country *j*, while $D_{ij}^\gamma$ represents the volume of trade between country *i* and the distance between countries *i* and *j*.

Since the model is nonlinear, we usually convert it to a linear model when studying the problem, and the logarithm of both parts of the equation takes the following form:

$$\ln T_{ij} = \beta_0 + \beta_1 \ln(Y_i Y_j) + \beta_2 \ln D_{ij} + \mu_{ij} \tag{2}$$

$\beta_0$, $\beta_1$, $\beta_2$ are the regression coefficients,
$\mu_{ij}$ is the standard random-error interference.

Forty countries that have signed the One Belt One Road agreement with China became the focus for the econometric study, namely: Russia, Vietnam, Indonesia, South Africa, Malaysia, South Korea, India, Thailand, Singapore, the Philippines, Turkey, Saudi Arabia, UAE, Greece, Brunei, Kazakhstan, Laos, Belarus, Lithuania, Hungary, Morocco, Ethiopia, Uganda, Egypt, Mongolia, Myanmar, Cambodia. Kuwait, Uzbekistan, Austria, Poland, Czech Republic, Slovakia, Portugal, New Zealand, Ecuador, Peru, Cuba, Chile, and Italy. All of these forty countries have signed cooperation documents with China for the joint creation of the "One Belt One Road" and have a large number of close trade exchanges with China, which facilitates the representativeness of the research results. The research indicators chosen in this paper include total bilateral trade, foreign direct investment [38], Ref. [110] GDP, GDP per capita, and geographic distance. In addition, two dummy variables are added: whether China has signed free trade agreements with its trading partners and whether the One Belt One Road initiative has been proposed. Furthermore, the following dummy variables are added: the COVID-19 pandemic and cross-country restrictions, and international sanctions against Russia.

Data on total bilateral trade and foreign direct investment come from the China Statistical Yearbook, and data on GDP and GDP per capita come from the official World Bank website [111]. Geographic distance is calculated as the distance between the capital of the two countries with data taken from Google Earth of China and its trading partner countries. Information about the existing signed free trade agreement is from the official website of the China Free Trade Zones [112].

*3.2. Factors Affecting the Gravity Model Application for Sino-Russian Trade*

Various factors affecting the practical application of the gravity trade model demonstrate that this model should not be taken as a given, but after a critical study of individual factors, it can provide useful empirical results.

The common border, language, and free trade area are so-called dummy variables, which means that if they are present, they take the value 1. If they are absent, they are 0.

Hampering factors/dummy variables include:

-    trade policy determinants such as tariffs, quotas, and subsidies;
-    differences between countries;
-    war; trade wars, intercountry restrictions

In addition to the above, considering the international situation, it is proposed for the purposes of analysis to supplement the dummy variables with such factors as international economic and other sanctions, the political situation, the COVID-19 pandemic, and the post-pandemic situation.

Supporting factors/dummy variables include:

-    bilateral and multilateral trade agreements;
-    the presence of trade organizations, such as the EFTA, NAFTA, and WTO;
-    similarities between countries or regions, such as the same native language of trading partners;
-    political system.

Sanctions enter the equation of gravity as trade costs when analyzing the consequences of their imposition [52,95,113–116]. The unobservable trade cost factor, $t_{ij}$, or bilateral trade resistance to sanctions with regard to the sanctions, is estimated as:

$$t_{ij}^k = b_{ij} d_{ij}^p \tag{3}$$

where $t_{ij}$ is a log-linear function of the observed two-way distance $d_{ij}$ and variable $b_{ij}$ variable that takes the value of 1, if $i$ and $j$ are in the same country, and, otherwise, 1 plus the tariff equivalent.

Multilateral resistance to sanctions affects relative prices. Therefore, their influence in this context can either lead to the destruction of foreign trade in extreme cases or have no effect. An intermediate option includes a deviation from existing trajectories, which can be defined as follows:

$$t_{ij}^k = b_{ij}d_{ij}^p\Pi_i = \left(\sum\left(\frac{t_{ij}}{P_j}\right)^{1-\sigma}\theta_i\right)^{1/(1-\sigma)} \tag{4}$$

where $\Pi_i$ is the "multilateral resistance" effect; $P_j$ measures the ease of access of importer $j$ to the market; $\theta_i$ is the price of bilateral trade.

The effect of sanctions is manifested if exports from country $i$ to country $j$ decrease due to increased trade costs $t_{ij}^k$. Trade variance is likewise caused by an increase $\Pi_i$. When multilateral resistance with country $j$ increases, the relative pressure with all other trading partners decreases. Therefore, the multilateral resistance effect of the sanctions affects these relative prices.

### 3.3. Approaches and Data Collection

This paper utilizes different approaches, such as grouping and generalization to summarize and generalize the broad references regarding the research hypothesis. Similarly, the methodology of scientific induction and deduction is performed out for providing fact-based proof of the hypothesis using the data. Moreover, the research methodology also uses different statistical techniques, such as descriptive statistics of trade indicators, for comparative and statistical analysis. Furthermore, and economical models such as the gravity model of international trade are used for the empirical analysis of the trade effect between Russia and China, and appropriate variables are introduced to analyze the main factors affecting the volume of Sino-Russian trade.

The study performed a quantitative analysis of the strengths and weaknesses of the competitiveness and complementarity of Russian and Chinese goods, allowing for problem analysis and recommendations. With regard to the analysis of the competitiveness of commodities, an explicit index of the comparative advantage [24] and a model of export commodity similarity index were selected to conduct a quantitative assessment of the competitiveness of Russian and Chinese commodities on different parameters. Regarding the commodity complementarity analysis between Russia and China, a rational quantitative review of the strength of commodity trade complementarity between the two dimensions is made by creating a commodity difference model and a comprehensive trade complementarity index model.

In relation to the main objectives and the aim of the study, a qualitative research approach is undertaken to investigate the research problem. The use of this approach is relevant and justified as it helps collect and analyze large, broader, and in-depth data array through either secondary sources or with the help of primary qualitative interviews. Moreover, qualitative data helps analyze the non-numerical data for understanding the experiences, opinions, and concepts and seeks a comprehensive understanding of phenomena in their natural setting [25]. Therefore, with the help of a qualitative approach, the development of the relationship between China and Russia is examined along with their impact on western countries. In contrast, the use of the quantitative approach, however, cannot be justified in this study as it relies on statistical and numerical analysis of the data, which is not required [117]. Although, the quantitative approach is useful in presenting a statistical analysis of the relationship and impact between the two variables, considering the scope of the study and the area of research (international politics), its application may not be suitable to collect and analyze the data [79]. Therefore, the qualitative approach is appropriately relevant and suitable for collecting the required data.

To analyze qualitative data, a thematic analysis technique is considered which is used for analyzing secondary data and is applied to a range of texts, including transcripts or interviews. In this study, the text is analyzed to examine the data and identify key ideas, and themes related to the research topic [118]. Hence, different themes are identified using thematic analysis based on the main objectives of the study and are analyzed accordingly with the support of relevant literature.

In this study, a secondary data source is used, which comprises books, newspapers, government reports, websites, journal articles, and personal sources. These types of were identified to be readily available and accessible compared to the primary source. Besides, it is less time-consuming and easy to collect than a primary source, which is a first-hand original source that requires time, and resources to gather information through different means and tools, largely from the research participants [119]. Therefore, the relationship between China and Russia is investigated through secondary sources, using scientific studies published in reputable journal articles, books, authentic and official government websites, and newspapers along with national reports. The study considers articles published online on international relations between China and Russia in areas including sustainable development, bilateral trade relationships, military relationships, and their impact on Europe and the US. The inclusion criteria comprise peer-reviewed journal articles, authentic government websites, and national reports publicly available and accessible in English only, since the year when the diplomatic relations between Russia and China were established. However, recent literature is emphasized by the latest data in the context of sanctions imposed on Russia.

The information and statistical base of the study was formed by the data of the Federal Customs Service of Russia, the Bank of Russia, Rosstat, the Eurasian Economic Commission (EEC), the United Nations (UN), the UN Conference on Trade and Development (UNCTAD), the WTO, the Organization for Economic Cooperation and Development (OECD), the International Monetary Fund (IMF), the World Bank, national banks, and statistical offices of the partner countries.

## 4. Results

### 4.1. Impact of the Sino-Russian Trade Relationship on Europe and the USA

The results were derived from the review of creditable secondary sources such as government reports, news, and extant literature. With thematic analysis used, the following themes were identified from the collected data via review of the extant literature and secondary data (Table 4).

**Table 4.** Themes for the data.

| Theme A: Sustainable Development of the Bilateral Trade Relationship between China and Russia | Theme B: Impact of the Sustainable Development of the Sino-Russian Trade Relationship on Europe and the USA under International Sanctions |
| --- | --- |
| Subthemes<br>Comparison of Foreign Policies and Interests;<br>Economic Ties and Bilateral Trade Relations | Subthemes<br>Economic Sanctions on Russia and the Impact on the USA and Europe;<br>Impact of Sino-Russian Relationships on the USA;<br>Impact of Sino-Russian Relationships on Europe |

#### 4.1.1. Economic Sanctions against Russia

As the Russia-Ukraine conflict emerged, many foreign companies that had previously invested in Russia withdrawn or suspended their activities for political and social reasons, domestic pressure, etc. or in response to the increasing difficulty of operating in Russia [36,120]. Consequently, Russia's economy has taken hit. In fact, it was predicted that the Russian economy would shrink by 8.5% by the end of 2022, the overall inflation would become all time high and reach 254%, while the unemployment rate would also increase up to 9.6 per cent [121]. The sanctions imposed by the USA and Europe were at the

forefront of these disruptions. These sanctions disrupted Russia's real economy–the production, sale, and transportation of goods. The state's trade volumes plummeted (Russian imports by volume were expected to fall by almost 25 per cent by the end of 2022) with the major international companies having reduced their operations, while some companies restricted their activities with Russia beyond what the sanctions legally require [36].

Analyzing the impact on the USA and Europe, it is worth noting that the USA does not have a significant economic relationship with Russia. Russia only made up close to 2.2% of the imports and exports of the USA [122]. Nonetheless, despite such low contribution of Russia to the US exports and imports, these sanctions have a significant negative implication for the US companies and sectors that have been operating in or with Russia. The concerns regarding the particular US financial institution's exposure to Russia can rise. So far, the major effect has been on the gasoline prices, which has exacerbated the inflation concerns and the energy crisis in the USA [123]. In the case of Europe, the sanctions have disrupted the global supply chain and contributed to the higher prices of various commodities [124]. Within the sanctions-imposing coalition, it is worth noting that the European Union has been the major economic partner of Russia. Europe has relied largely on the supply of energy from Russia; it sources its natural gas from Russia. In this regard, Russia is seeking to deepen economic ties with countries outside the sanction's coalition such as China.

### 4.1.2. Impact of Sino-Russian Relationships on the USA

It is observed that Russia and China are being closer to each other day by day. The Sino-Russian relationship coordination on different operations can have a significant impact on other countries. Due to increasing the ties between both countries, the US is also changing its behavior and policies toward Russia [125]. Russia and China collaboratively can have a severe long-term negative impact on the US economy as according to the regulatory authorities of the US, the coordination and association between Russia and China are just to undermine the current international order. It can also bring several challenges with respect to the strategic areas of the US; however, these challenges were forecasted earlier by most policy makers in the US [126]. In addition to this, it has been found that the Sino-Russian partnership also contains uncertainty about the continuity and sustainability of the operations as both countries face significant challenges that can have a positive impact on Russia. The partnership and coordination between both of these countries are encouraged by sober appreciation, and one of the main challenges is that both countries have distinct agendas and goals. The United States, Russia, and China are the three major economies, and the overall triangle of the Sino-Russian-US relationships had gradually shifted toward strong Sino-Russian relations during the last three decades, which increased the concerns of the United States [127].

Moreover, the world is seeing China as a growing superpower as it is containing the world's largest economy. In this regard, increasing coordination and participation between China and Russia is considered the biggest problem specifically for the US. The majority of the resident and policymakers of the US believed that after a few years, the United will to lose its dominancy in the different regions of the world [128]. This is just because of the current ineffective international policies of the government. According to them, their policymakers failed to plan their strategies proactively and were results-driven. It has also been found that most Americans believe that increasing Sino-Russia cooperation is the result of China's good policies as they believe that there is only one way of succeeding eventually, which is the economic and strategic corporation. Most Americans think that increasing China's corporation with most of the world's leading countries is one of the biggest threats to America [129]. Additionally, it was also found that Russia and China have different histories of economic and regional growth but contain similarities in interest. Their relationship and cooperation in different sectors are specifically driven by the alignment of their interest. On the other hand, the policies of the Western countries (specifically the United States) are not supported and are in the favor of Russia, which is providing a

competitive edge to China, compared to the US. Hence, if the processes continually grow with the same pace, it can have long-lasting economic consequences for the US [130].

### 4.1.3. Impact of Sino-Russian Relationships on Europe

Europe considers Russia and China major allies as both countries play a significant role in the provision of different commodities and exchange of different products. Russia sees Europe as complementary to the Chinese market. This means that Chinese exports are the closest substitute to Europe exports specifically in Russian markets. The results suggest that the increase in the Sino-Russian relationship can have a significant and negative impact on the overall exports of Europe [131]. China is in the consistent phase of its development, and continuity exists in the overall increment of its exports. China has developed its expertise in the different sectors of their industry, the efficient production and better quality of the products, including advanced machinery attracting different countries to create long-term trade relationships with China [132]. The enhancement in the Sino-Russian relationship can negatively impact the different industries in Europe. The most affected European industries due to the development of the Sino-Russian relationship include equipment and machinery, nuclear reactors, and electronic machinery. Additionally, Europe is also trying to compete with China by developing new technologies and bringing advancements in their operation to produce high-quality products at relatively affordable costs. Therefore, this is a healthy competition as Europe is consistently aiming to upgrade its industry [133].

China has a competitive edge over Europe in terms of the factors of production. The cost of land, labor, capital, and raw materials is slightly higher in the European region, which negatively influences convincing the other countries to create long-term trade relationships. Moreover, the importance of the Sino-Russian relationship can be increased once the Belt and Road initiative of China is successfully completed [134]. The impacts of the Sino-Russian cooperation can be divided into two different scenarios, including the short-term and the long-run impacts. In the short run, it is challenging for European countries to enhance their exports as their competitors (China) has comparative advantages over them. Eventually, it can bring several opportunities for European countries. Eventually, their close ties and relationships will be helpful for the European countries in the expansion of trading activities and in the provision of access to new markets. The Belt and Road initiative facilitates European countries to reduce their transportation cost and access to link different regions of the work via Russia. Hence, in the short run, the Sino-Russian relationship is negatively impacting European countries in terms of the reduction of their exports. However, it is facilitating European countries to develop their expertise to compete with the world, which consequently improves their economic conditions [135].

### 4.2. Analysis of the Competitiveness of Russian-Chinese Trade in Goods

The analysis of the index of revealed comparative advantage is a widely used indicator to measure the strength of competitiveness and complementarity, which uses the single classification of goods, based on the classification of data in this paper, for the calculation [136]. Single classification is a method of classifying goods in SITC Rev.3 across broad categories from 0 to 9, without expanding the division of goods into each category. The single classification allows one to observe aggregate data for each commodity group, eliminating the need to calculate the subdivision for each commodity group and helping simplify calculations and capture the aggregate.

Revealed Comparative Advantage (RCA) is the ratio between the share of a particular commodity exports in a country's total exports and that commodity share in total world exports:

*RCA* > 1 means that the country has a significant comparative advantage in that commodity;

*RCA* < 1 means that the country does not have a significant comparative advantage in that commodity.

$RCA_{xm}^n$ is the dominant comparative advantage of country $m$ for product category $n$;
$X_m^n$ is the export value of the commodity category $n$ in country $m$;
$X_m$ is the total export value of all commodity categories from 0 to 9 in country $m$;
$X_w^n$ is the total export value of the commodity category $n$ in the world;
$X_w$ is the total export value of all product categories from 0 to 9 in the world [136].

As shown in Table 5, among China's exports to the world market, the indices of the dominant comparative advantage for various goods (SITC8), cars and transport equipment (SITC7), and industrial raw materials (SITC6) are greater than 1, which means the dominant comparative advantage, while the indices of dominant comparative advantage for other categories are less than 1, which means no comparative advantage. The index of significant comparative advantage for various industrial goods (SITC8) ranges from 1.8 to 3, with the most obvious significant comparative advantage; the index of significant comparative advantage for cars and transport equipment (SITC7) slowly declines, ranging from 1.26 to 1.46, and the index of a significant comparative advantage for industrial raw materials (SITC6) has a stable overall trend, remaining at about 1.35, with a weaker significant comparative advantage. The dominant comparative advantage was poor.

**Table 5.** Index of the clear comparative advantages of various Chinese products from 2008 to 2022, calculated by the authors based on data of [14,137].

| Year | SITC0 | SITC1 | SITC2 | SITC3 | SITC4 | SITC5 | SITC6 | SITC7 | SITC8 | SITC9 |
|------|-------|-------|-------|-------|-------|-------|-------|-------|-------|-------|
| 2008 | 0.42 | 0.13 | 0.23 | 0.16 | 0.12 | 0.51 | 1.32 | 1.30 | 2.16 | 0.03 |
| 2009 | 0.44 | 0.15 | 0.20 | 0.15 | 0.07 | 0.44 | 1.22 | 1.41 | 2.10 | 0.03 |
| 2010 | 0.46 | 0.16 | 0.18 | 0.13 | 0.05 | 0.49 | 1.23 | 1.42 | 2.14 | 0.02 |
| 2011 | 0.46 | 0.16 | 0.18 | 0.11 | 0.05 | 0.56 | 1.30 | 1.46 | 2.27 | 0.03 |
| 2012 | 0.44 | 0.16 | 0.17 | 0.10 | 0.05 | 0.51 | 1.30 | 1.42 | 2.32 | 0.01 |
| 2013 | 0.42 | 0.15 | 0.17 | 0.09 | 0.05 | 0.51 | 1.34 | 1.45 | 2.35 | 0.01 |
| 2014 | 0.41 | 0.16 | 0.18 | 0.10 | 0.06 | 0.53 | 1.37 | 1.36 | 2.27 | 0.02 |
| 2015 | 0.40 | 0.17 | 0.18 | 0.12 | 0.06 | 0.51 | 1.36 | 1.27 | 2.01 | 0.02 |
| 2016 | 0.44 | 0.19 | 0.18 | 0.14 | 0.05 | 0.51 | 1.35 | 1.26 | 2.99 | 0.05 |
| 2017 | 0.43 | 0.18 | 0.17 | 0.17 | 0.07 | 0.55 | 1.32 | 1.30 | 2.00 | 0.04 |
| 2018 | 0.43 | 0.18 | 0.19 | 0.18 | 0.09 | 0.58 | 1.33 | 1.33 | 2.97 | 0.04 |
| 2019 | 0.41 | 0.16 | 0.18 | 0.19 | 0.10 | 0.56 | 1.37 | 1.30 | 1.92 | 0.09 |
| 2020 | 0.36 | 0.11 | 0.15 | 0.17 | 0.09 | 0.53 | 1.38 | 1.29 | 1.86 | 0.19 |
| 2021 | 0.41 | 0.14 | 0.19 | 0.18 | 0.11 | 0.55 | 1.39 | 1.32 | 2.36 | 0.09 |
| 2022 | 0.43 | 0.17 | 0.18 | 0.19 | 0.09 | 0.54 | 1.37 | 1.46 | 2.01 | 0.12 |

Table 6 shows that among goods exported by Russia to the world market, crude oil and minerals (SITC3) have a significant comparative advantage, while the index of a significant comparative advantage for crude oil and minerals (SITC3) is well above 4, even peaked at 6.31 in 2015, and exceeded 4.9 for five consecutive years after that, which shows a clear significant comparative advantage. From 2012 to 2015, SITC9 had no significant comparative advantage, but after 2016, the SITC9 Index of the Significant Comparative Advantage became greater than 2 and has maintained steady growth, showing a comparative advantage. The dominant comparative advantage index for the remaining product groups fluctuates around 1, with little comparative advantage. The remaining product groups have no comparative advantage, with a clear comparative advantage index of less than 1.

**Table 6.** The index of clear comparative advantages of various goods in Russia from 2008 to 2022 was calculated by the authors based on data of [14,137].

| Year | SITC0 | SITC1 | SITC2 | SITC3 | SITC4 | SITC5 | SITC6 | SITC7 | SITC8 | SITC9 |
|------|-------|-------|-------|-------|-------|-------|-------|-------|-------|-------|
| 2008 | 0.26 | 0.24 | 1.03 | 4.81 | 0.50 | 0.44 | 0.87 | 0.09 | 0.06 | 1.83 |
| 2009 | 0.40 | 0.30 | 0.91 | 5.49 | 0.66 | 0.35 | 0.97 | 0.10 | 0.06 | 2.01 |
| 2010 | 0.28 | 0.20 | 0.77 | 4.98 | 0.30 | 0.36 | 0.87 | 0.08 | 0.05 | 2.40 |
| 2011 | 0.32 | 0.19 | 0.76 | 4.24 | 0.32 | 0.39 | 0.76 | 0.07 | 0.04 | 2.46 |
| 2012 | 0.43 | 0.28 | 0.78 | 4.54 | 0.70 | 0.44 | 0.87 | 0.11 | 0.08 | 0.61 |
| 2013 | 0.40 | 0.30 | 0.78 | 4.34 | 0.77 | 0.42 | 0.84 | 0.12 | 0.10 | 0.59 |
| 2014 | 0.49 | 0.33 | 0.86 | 4.59 | 0.87 | 0.44 | 0.85 | 0.12 | 0.11 | 0.58 |
| 2015 | 0.58 | 0.40 | 1.02 | 6.31 | 1.02 | 0.52 | 1.00 | 0.15 | 0.13 | 0.65 |
| 2016 | 0.67 | 0.41 | 1.20 | 5.46 | 1.32 | 0.51 | 0.17 | 0.17 | 0.17 | 2.33 |
| 2017 | 0.67 | 0.35 | 1.13 | 5.20 | 1.25 | 0.49 | 1.11 | 0.16 | 0.12 | 2.35 |
| 2018 | 0.72 | 0.27 | 1.11 | 4.90 | 1.11 | 0.43 | 1.07 | 0.13 | 0.09 | 2.19 |
| 2019 | 0.70 | 0.30 | 1.14 | 5.12 | 1.56 | 0.44 | 1.04 | 0.14 | 0.11 | 2.40 |
| 2020 | 0.92 | 0.38 | 1.32 | 5.70 | 1.86 | 0.46 | 1.28 | 0.13 | 0.12 | 2.88 |
| 2021 | 0.90 | 0.28 | 1.31 | 5.81 | 1.87 | 0.50 | 1.19 | 0.12 | 0.11 | 2.81 |
| 2022 | 0.93 | 0.31 | 1.32 | 5.87 | 1.89 | 0.52 | 1.29 | 0.14 | 0.14 | 2.98 |

According to Tables 3 and 4, the analysis of the Russia-China index of the clear comparative advantage in the world market leads to the following conclusions: China's comparative advantage in the world market is in the categories of industrial raw materials (SITC6), cars and transport equipment (SITC7) and various industrial goods (SITC8), while Russia's comparative advantage in the world market is in the categories of industrial raw materials (SITC2), crude oil and minerals (SITC3), animal and vegetable oils and fats (SITC4), industrial raw materials (SITC6) and other goods (SITC9).

Analysis of Russian and Chinese import and export data for the time span from 2008 to 2022 revealed three product categories in which China has a clear comparative advantage, accounting for about 89% of all Russian imports to China, and four product categories in which Russia has a clear comparative advantage, accounting for about 74.6% of all Chinese imports to Russia. The product categories in which Russia and China have a clear comparative advantage do not overlap, meaning that Russia and China strongly complement each other in commodity trade.

As for the description of trade competitiveness between the two countries, we have chosen the export similarity index model.

$$\text{ESI}\,(ab,\,c) = \sum_i \left[ \frac{\left(\frac{X_{ac}^i}{X_{ac}}\right) + \left(\frac{X_{bc}^i}{X_{bc}}\right)}{2} \right] \cdot \left[ 1 - \left| \frac{\left(\frac{X_{ac}^i}{X_{ac}}\right) - \left(\frac{X_{bc}^i}{X_{bc}}\right)}{\left(\frac{X_{ac}^i}{X_{ac}}\right) + \left(\frac{X_{bc}^i}{X_{bc}}\right)} \right| \right] \times 100\% \tag{5}$$

ESI $(ab,\,c)$ is the export similarity index between exports from country $a$ and country $b$ to market $c$,

$a$, $b$ are the two countries in question, respectively,

$c$ is a world market.

$X$ is export,

$\frac{X_{ac}^i}{X_{ac}}$ is the export of product category $i$ from country $a$ to market $c$,

$X_{ac}$ is the export of all goods from country $a$ to market $c$,

$\frac{X_{bc}^i}{X_{bc}}$ is the export of product category $i$ from country $b$ to market $c$,

$X_{bc}$ is the export of all goods from country $b$ to market $c$.

The export similarity index ranges from 0 to 1, with a value closer to 0 indicating that the two countries' exports are less competitive in the world market, and closer to 1 indicating that the two countries' goods are more competitive in the world market [138].

As shown in Table 7 and Figure 5, the similarity index between China and Russia's exports to the world market from 2008 to 2022 is low, ranging from 0.21 to 0.35 and remaining around 0.27 on average, indicating that China and Russia's exports to the world market are very uncompetitive.

**Table 7.** Similarity index between Russian and Chinese exports 2008–2022; developed by authors based on certain data from [14,137].

| Year | 2008 | 2009 | 2010 | 2011 | 2012 | 2013 | 2014 | 2015 | 2016 | 2017 | 2018 | 2019 | 2020 | 2021 | 2022 |
|---|---|---|---|---|---|---|---|---|---|---|---|---|---|---|---|
| I* | 0.255 | 0.26 | 0.228 | 0.213 | 0.249 | 0.246 | 0.255 | 0.301 | 0.344 | 0.32 | 0.294 | 0.298 | 0.332 | 0.297 | 0.35 |

I*—Index of similarity of export goods.

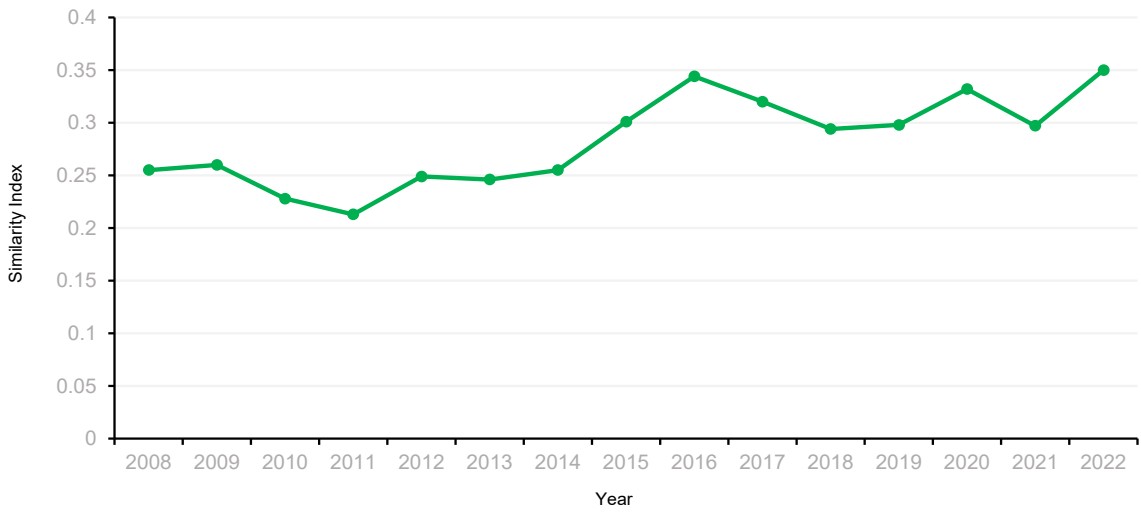

**Figure 5.** Similarity Index Diagram for Russian and Chinese exports; developed by authors based on certain data from [14,137].

### 4.3. Analysis of the Complexity of Sino-Russian Trade

We choose the product differentiation analysis method and the composite trade complementarity index model to analyze the Russian-Chinese trade complementarity, in which the composite trade complementarity index is classified according to the commodity data, and the unambiguous classification commodity data is used as the data source.

The trade differentiation between the two countries is usually measured by the product differentiation index (PD) [111].

$$PD_{ij}^k = \frac{\left| X_{ij}^k - M_{ij}^k \right|}{X_{ij}^k + M_{ij}^k} \tag{6}$$

$PD_{ij}^k$ is the differentiation index of $k$ commodities between countries $i$ and $j$;

$X_{ij}^k$ is the cost of exporting $k$ commodities from country $i$ to country $j$;

$M_{ij}^k$ is the cost of importing $k$ commodities from country $i$ to country $j$.

$0 < PD_{ij}^k < 1$, closer $PD_{ij}^k$ to 1, the higher the degree of differentiation $k$ of traded goods between the two countries; the more $PD_{ij}^k$ tends to 0, the less differentiated the two countries are in terms of the $k$ commodities traded in total trade between the two countries.

$$PD = \sum_k \left( PD_{ij}^k \times \frac{X_{ij}^k + M_{ij}^k}{X_{ij} + M_{ij}} \right) \tag{7}$$

$PD$ is the total index of product differentiation in trade between the two countries.

$PD_{ij}^k$ is the index of product differentiation of product category $k$ in trade between countries $i$ and $j$,

$M_{ij}^k$ is the cost of importing product category $k$ from country $i$ to country $j$,

$X_{ij}$ is the total value of exports from country $i$ to country $j$,

$M_{ij}$ is the total value of imports from country $i$ to country $j$.

The higher the PD index, the higher the degree of product differentiation between the two countries; the lower the PD index, the lower the degree of product differentiation between the two countries.

When the PD index is greater than 0 and less than 0.25 means low product differentiation in total trade between the two countries, being greater than 0.25 and less than 0.5 means low product differentiation in total trade between the two countries, being greater than 0.5 and less than 0.75 means high product differentiation in total trade between the two countries, and being greater than 0.75 and less than or equal to 1 means high product differentiation in total trade between the two countries.

As shown in Table 8 and Figure 6, the product differentiation index for Sino-Russian trade ranges from 0.69 to 0.86, as a whole, from 2008 to 2022, which is in the higher range of differentiation. The product differentiation index for Russian-Chinese trade fluctuates but remains consistently high, with an average value of 0.80. This suggests that the product structure of Russian-Chinese trade is more differentiated, less competitive, and more complementary and has shown an upward trend recently.

**Table 8.** The index of product differentiation in Sino-Russian trade; developed by authors based on certain data from [14,137].

| Year | 2008 | 2009 | 2010 | 2011 | 2012 | 2013 | 2014 | 2015 | 2016 | 2017 | 2018 | 2019 | 2020 | 2021 | 2022 |
|------|------|------|------|------|------|------|------|------|------|------|------|------|------|------|------|
| I* | 0.804 | 0.693 | 0.754 | 0.788 | 0.832 | 0.839 | 0.860 | 0.767 | 0.787 | 0.839 | 0.845 | 0.828 | 0.767 | 0.801 | 0.832 |

I*—Trade products difference index.

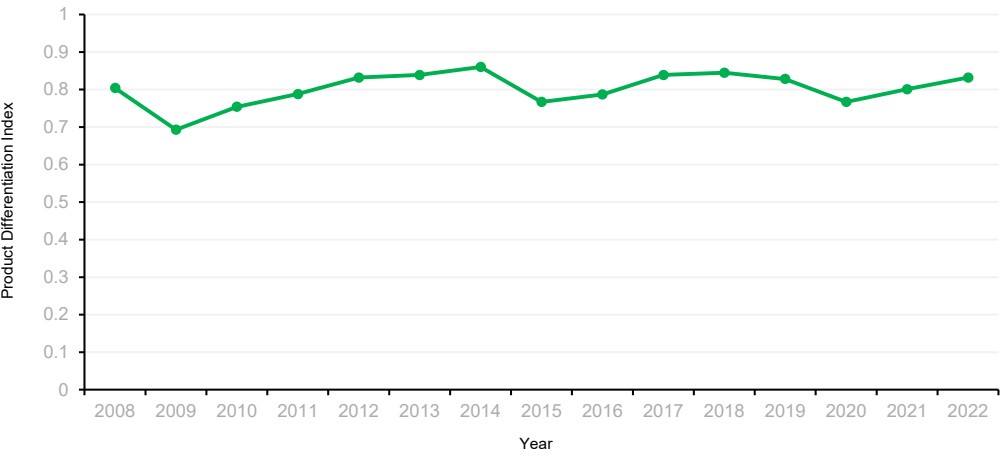

**Figure 6.** The index of product differentiation in trade between Russia and China; developed by authors based on certain data from [137].

The Trade Complementarity Composite Index (TCI) is an index model used to measure the complementarity of export and import trade between the two trading countries that interact with each other [111]. This relies on the following formula:

$$TC_{2j} = \sum_k \left[ \left( RCA_{kj}^k \times RCA_{mj}^k \right) \times \left( X_w^k / X_w \right) \right] \tag{8}$$

$RCA_{kj}^k$ is the index of the dominant comparative advantage for product category $k$ in country $i$;

$RCA_{mj}^{k}$ is the dominant comparative disadvantage for product category *k* in country *j*;

$X_{w}^{k}$ denotes the total exports of product category *k* in the world; *X* denotes the total exports of all ten product categories globally.

The explicit index of comparative disadvantages [111] is calculated using the following formula:

$$RCA_{mj}^{k} = \frac{M_{j}^{k}/M_{j}}{X_{w}^{k}/X_{w}} \tag{9}$$

$M_{j}^{k}$ is the import of country *j* for product category *k*;

$M_{j}$ is the total import of all ten categories of goods in the country *j*;

$X_{w}^{k}$ denotes the total exports of the commodity category *k* globally;

$X_{w}$ denotes the total exports of all ten categories of goods globally.

If the composite trade complementarity index is greater than 1, it suggests that there is greater import and export trade complementarity between the two countries in product category *k*. A larger composite trade complementarity index means that the greater the complementarity between the import and export trade of the two countries, the better a country's overall export structure matches that of its trading country, and the more likely that trade between the two countries will promote each other and the welfare of the two countries [139].

Examining the Relative Comparative Advantage (RCA) is important in understanding the potential for trade and economic cooperation between the two countries and identifying areas where each country can specialize and benefit from trade. Furthermore, it helps to identify the sectors in which a country has a comparative advantage and can compete globally. This knowledge is vital for policymakers, as it can inform the development of trade policies and the allocation of resources to support the growth of these sectors. By focusing on areas of the mutual advantage, the two countries can benefit from economies of scale and reduce production costs, making them more attractive to potential trading partners. Ultimately, examining RCA can facilitate the growth of trade and economic cooperation between countries, leading to increased economic development and competitiveness.

Table 9 demonstrates that the complementarity of commodity trade between China and Russia is strong, and Figure 7 shows a zigzag upward trend from 2008 to 2022. Generally, when China is a commodity exporter and Russia is a commodity importer, the overall trade complementarity index between China and Russia tends to slowly decline, reaching a maximum of about 1.23 in 2012 and 2013 and a minimum of about 1.07 in 2015, but it is always above 1 and the average value remains around 1.17, indicating that the overall import and export trade between China and Russia has strong complementarity under these conditions. When Russia is a commodity exporter and China is an importer, the China-Russia trade complementarity index shows an upward trend, increasing from about 0.94 in 2011 to about 1.175 in 2018, an increase of more than 33%, while the average value remains at 1.09, indicating that China-Russia import-export trade has strong complementarity under such conditions. Overall, there is strong complementarity between China and Russia's commodity trade, which is closely related to the current situation of China's rapid industrial development and Russia's basic position as an energy power.

**Table 9.** Aggregate index of trade in goods between Russia and China; calculated by authors based on [14,137].

| Year | Chinese Export to Russia | Russian Exports to China |
|------|--------------------------|--------------------------|
| 2008 | 1.083 | 0.968 |
| 2009 | 1.079 | 0.993 |
| 2010 | 1.116 | 0.966 |
| 2011 | 1.121 | 0.946 |
| 2012 | 1.239 | 1.030 |
| 2013 | 1.237 | 0.989 |
| 2014 | 1.184 | 1.057 |
| 2015 | 1.071 | 1.116 |
| 2016 | 1.118 | 1.108 |
| 2017 | 1.141 | 1.189 |
| 2018 | 1.376 | 1.176 |
| 2019 | 1.245 | 1.187 |
| 2020 | 1.212 | 1.175 |
| 2021 | 1.781 | 1.90 |
| 2022 | 1.971 | 2.01 |

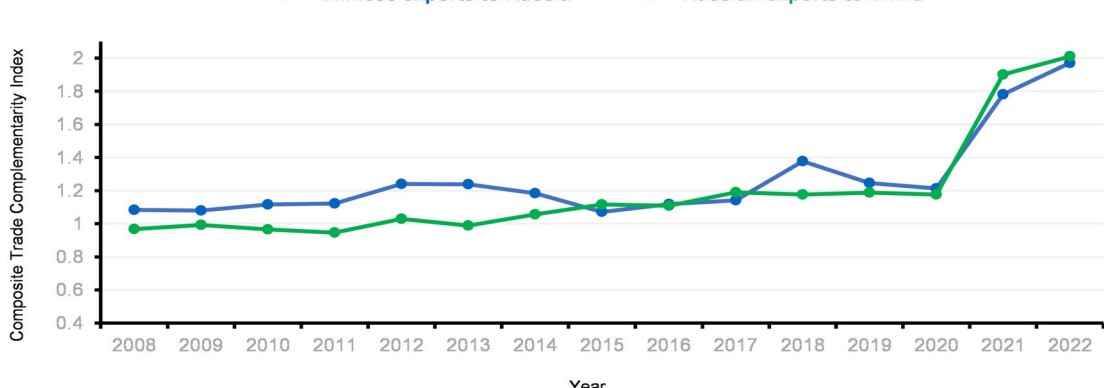

**Figure 7.** Diagram of the aggregate Russia-China trade index for commodities; developed by authors based on certain data from [137].

According to the empirical analysis carried out during the period from 2008 to 2022 and based on actual data on the scale and structure of Russian-Chinese trade and using a multidimensional index model, we can conclude that this bilateral trade is characterized more by complementarity than competition, and its complementarity is the main characteristic.

### 4.4. Current Problems of Russian-Chinese Trade Development

Russia and China have different economic development processes and different economic structures; the basic economic conditions of the two countries make bilateral trade a great potential for complementary trade, while both parties export goods to the world market, and bilateral import-export trade has very little competition, which provides the basis for the stable development of bilateral trade between Russia and China. However, the Russia-China trade structure is overly dependent on a certain type of commodity, as the basic structure of commodity exports remains unchanged. However, if there is an unfavorable change in the prices of the leading commodities, which account for more than 50% of the export structure, it will have a direct impact on Russia-China trade and create more uncertainty for the bilateral trade of these countries.

The most obvious problem is the structure of Russian exports to China, where the bulk of exports are crude oil and minerals, averaging over 50% and, in some years, over

70%. Although global oil prices have begun to recover since 2017, medium- and long-term forecasts from both the World Bank and OPEC suggest that oil prices will not exceed $80 per barrel between 2020 and 2025, and Russia's overall foreign trade with China will be affected by overreliance on oil prices [106]. Although Russia has prioritized the structural transformation of its economy since 2014, measures taken by the Russian government as a whole have had little short-term impact on changing the structure of Russian foreign trade exports. The need to diversify Russia's economic development still faces strong structural constraints and challenges to a sustainable foreign trade model. This export structure, which is overly dependent on certain types of products, makes Russian-Chinese bilateral trade vulnerable to adverse international shocks and increases trade risks [14].

Russia-China bilateral trade is highly complementary, but according to the analysis of the actual scale and growth rate of trade between these countries, Russia-China trade has not yet fully benefited from it. Compared to the volume and growth rate of trade between China-the United States and China-South Korea over the same period, there is enormous potential for developing trade between Russia and China.

The development of cross-border e-commerce cooperation between Russia and China has lagged behind. In particular, no specialized platform has been created for Russia to develop cross-border e-commerce, which to a certain extent hinders the use of the additional potential of Russian-Chinese trade.

The brand value and competitiveness of Chinese products are low. First, when the Russian market opened in 1991, Chinese consumer goods dominated it. Only at the beginning of this century, the rapid technological development of China led to technical goods taking up the bulk of Chinese exports to Russia, but light industrial goods still hold a certain share. The emphasis on the brand value of Chinese goods is not obvious in relation to the inherent interchangeability of these goods that leads to low brand recognition of Chinese industrial goods in the Russian market, which weakens the competitiveness of Chinese goods.

### 4.5. An Empirical Analysis of the Effects of Russian-Chinese Trade

Through the analysis of the original model of trade gravity, according to the selected indicators, the new model is completed by authors and adapted to the modern realities based on panel data:

$$\ln(T_{ijt}) = \beta_0 + \beta_1 \ln(Y_{it} Y_{jt}) + \beta_2 \ln(I_{ijt}) + \beta_3 \ln(GDP_{ijt}) + \beta_4 \ln(D_{ij}) + \beta_5 F_{ijt} + \beta_6 B_j + \mu itj + \beta_7 Cov jt + \beta_8 S jt \tag{10}$$

$i$ denotes China;

$j$ denotes Russia;

$t$ represents the year;

$T_{ijt}$ is the total volume of bilateral trade between China $i$ and Russia $j$ in year $t$. The greater the volume, the closer the trade;

$Y_{it}$ is China's GDP in year $t$;

$Y_{jt}$ is the trade partner's GDP in year $t$, Representing the economic aggregate of the two countries;

$I_{ijt}$ is the amount of direct investment of Russia $j$ in China $i$ in year $t$, and the investment that can stimulate economic and trade growth;

$GDPijt$ is the indicator that shows the absolute value of the difference in GDP per capita between China $i$ and Russia $j$ in year $t$. The greater the difference in factor endowment between the two countries, the greater the inter-industry trade between the two countries and the smaller the volume of bilateral trade;

$D_{ij}$ is the geographical distance between the two countries. The greater the distance, the higher the cost of trade, and the lower the volume of bilateral trade;

$F_{ijt}$ is a dummy variable indicating whether China and Russia signed a free trade agreement. If "yes", it is assigned a value of 1, otherwise, it is assigned a value of 0,

$Bj$ is a dummy variable indicating the country's inclusion in the *One Way–One Belt* policy.

Cov*jt* is a dummy variable showing the impact of the COVID-19 pandemic on trading partner *j* in year *t*. If yes, it is assigned the value of 1; otherwise, it is assigned the value of 0.

S*jt* is a dummy variable showing the impact of international sanctions on trading partner *j* in year *t*. If yes, it is assigned the value of 1; otherwise, it is assigned the value of 0.

### 4.5.1. The Unit Root Test

To avoid the phenomenon of "pseudo-regression" in the model and to ensure the accuracy of the empirical analysis, one must first perform a unit root test on the model. The unit root test method for panel data is similar, but not identical to the unit root test for normal time series. Many methods can be used for testing the unit root, but the authors used the two most basic and most commonly used unit root testing methods, namely the LLC test and the Fisher-ADF test. Two dummy variables and an independent distance variable of distance between two countries that do not change over time are removed, and a unit root test is performed for the other variables. See the results in Tables 10 and 11.

**Table 10.** Test results of the LLC test panel root data block; developed by authors based on certain data from [14,137].

| Variable | The Method Check | Statistics LLC | *p*-Value | The Result |
|---|---|---|---|---|
| $\ln(\ddot{O}_{ijt})$ | (C,0,K) | −14.9369 | 0.0000 | stationary |
| $\ln(T_{i\text{榛}t})$ | (C,T,K) | −19.4627 | 0.0000 | stationary |
| $\ln(T_{ijt})$ | (0,0,K) | 10.1566 | 1.0000 | nonstationary |
| $\ln(Y_{it}Y_{ij})$ | (C,0,K) | −20.7800 | 0.0000 | stationary |
| $\ln(Y_{it}Y_{ij})$ | (C,T,K) | −16.3693 | 0.0000 | stationary |
| $\ln(Y_{it}Y_{ij})$ | (0,0,K) | 8.38516 | 1.0000 | nonstationary |
| $\ln(I_{ijt})$ | (C,0,K) | −2.28841 | 0.0111 | stationary |
| $\ln(I_{ijt})$ | (C,T,K) | −2.83301 | 0.0023 | stationary |
| $\ln(I_{ijt})$ | (0,0,K) | −1.37935 | 0.0839 | nonstationary |
| $\ln(GDP_{ijt})$ | (C,0,K) | −5.30851 | 0.0000 | stationary |
| $\ln(GDP_{ijt})$ | (C,T,K) | −12.3794 | 0.0000 | stationary |
| $\ln(GDP_{ijt})$ | (0,0,K) | 0.97663 | 0.8356 | nonstationary |

Note: (C,T,K) C in brackets means LLC criterion with constant term (C = 0 means no constant term), T means trend term (T = 0 means no trend term), K means lag order, and the maximum lag value is automatically chosen according to the Schwarz criterion.

**Table 11.** Test results of the test panel root data block Fisher-ADF; developed by authors based on certain data from [14,137].

| Variable | The Method Check | Statistics Fisher-AD | *p*-Value | The Result |
|---|---|---|---|---|
| $\ln(T_{ijt})$ | (C,0,K) | 213.552 | 0.0000 | stationary |
| $\ln(T_{ijt})$ | (C,T,K) | 209.429 | 0.0000 | stationary |
| $\ln(T_{ijt})$ | (0,0,K) | 9.08600 | 1.0000 | nonstationary |
| $\ln(Y_{it}Y_{ij})$ | (C,0,K) | 275.369 | 0.0000 | stationary |
| $\ln(Y_{it}Y_{ij})$ | (C,T,K) | 164.844 | 0.0000 | stationary |
| $\ln(Y_{it}Y_{ij})$ | (0,0,K) | 7.57311 | 1.0000 | nonstationary |
| $\ln(I_{ijt})$ | (C,0,K) | 110.102 | 0.0145 | stationary |
| $\ln(I_{ijt})$ | (C,T,K) | 94.3436 | 0.1304 | nonstationary |
| $\ln(I_{ijt})$ | (0,0,K) | 119.117 | 0.0030 | stationary |
| $\ln(GDP_{ijt})$ | (C,0,K) | 71.4636 | 0.7413 | nonstationary |
| $\ln(GDP_{ijt})$ | (C,T,K) | 123.066 | 0.0014 | stationary |
| $\ln(GDP_{ijt})$ | (0,0,K) | 67.5869 | 0.8374 | nonstationary |

Note: (C,T,K) C in brackets means LLC criterion with constant term (C = 0 means no constant term), T means trend term (T = 0 means no trend term), K means lag order, and the maximum lag value is automatically selected according to the Schwarz criterion.

All the above panel data passed the unit root test and a stable time series, which avoids the phenomenon of "pseudo-regression" and makes it possible to regress the model.

### 4.5.2. Hausman Test

In the Hausman test, if the $p$ value is greater than 0.1, the null hypothesis is accepted and the model with random effects is chosen; if the $p$ value is less than 0.1, the null hypothesis is rejected and the model with fixed effects is chosen [140]. The specific results are shown in Table 12.

**Table 12.** The results of the Hausman test.

| Test Summary Chi | Chi Sq. Statistic | Chi Sq. d.f. | Prob. |
|---|---|---|---|
| Cross-section random | 19.506789 | 4 | 0.5448 |

According to the output results, the value of the Hausman statistic is 19.506789 and the corresponding $p$ value is 0.5448, which means that the initial hypothesis is valid, and the random effects model works out.

### 4.5.3. Regression Model

With selected panel data for China from 2008 to 2020 [14,137], the regression of sample data was performed using the econometric software *Eviews11* based on the above test results.

The first regression results were as follows:

$$
\begin{aligned}
\ln(T_{ijt}) &= 0.685989 \ln(Y_{it} Y_{ij}) \\
&+ 0.36289l(I_{ijt}) + 0.018221 \ln(GDP_{ijt}) - 0.595220 \ln(D_{ij}) \\
&+ 0.225142 F_{ijt} + 0.108602 B_j - 3.385354 \\
&(19.84025)(0.500760)(0.919959)(-3.666964)(3.458057)(2.741776)(-2.148902)
\end{aligned}
\tag{11}
$$

$$
R^2 = 0.631573 \ \overline{R}^2 = 0.6272264 \ F = 146.5679
$$

The regression coefficient for the explanatory variable $\ln(GDP_{ijt})$ is positive, which does not correspond to the expected sign. The reason for this is that there is multicollinearity between the independent variables, indicating a strong linear correlation between the explanatory variable $\ln(GDP_{ijt})$ and other independent variables. To make the overall effect of the equation fitting more ideal, the explanatory variable $\ln(GDP_{ijt})$ is out.

For the second regression, the specific regression results were as follows:

$$
\begin{aligned}
\ln(T_{ijt}) &= 0.692962 \ln(Y_{it} Y_{ij}) \\
&+ 0.003874l(I_{ijt}) - 0.601511 \ln(D_{ij}) + 0.177371 F_{ijt} \\
&+ 0.112503 B_j - 3.299611 \\
&(20.56656)(0.534273)(-3.684473)(1.430016)(2.854868)(-2.084801)
\end{aligned}
\tag{12}
$$

$$
R^2 = 0.630682 \ \overline{R}^2 = 0.627089 \ F = 175.5507
$$

All signs of the coefficients of independent variables in the regression results correspond to the expected results.

### 4.6. Analysis of the Results of the Empirical Regression

According to the above equation model, the regression results are analyzed, and each index is sorted by the degree of influence on the volume of bilateral trade between China and Russia as follows: GDP of China and Russia > distance between China and Russia > whether China and Russia have signed a free trade agreement > whether the One Belt One Road initiative is proposed > foreign direct investment (coefficient is positive).

Based on the results of econometric modeling, the following conclusions can be drawn:

1. GDP of China and Russia play an important role in increasing the volume of bilateral trade (0.692962). The coefficient of the product of the explanatory variables of the GDP of the two countries is 0.692962, which means that for every 1% increase in the logarithm of the product of the GDP of the two countries, the logarithm of the volume of bilateral trade between the two countries will increase by 0.692962%. This also shows that the increase in GDP, the increase in the economic aggregate and economic scale of China and Russia, and the improvement of the economic development level of these countries play an important role in the development of bilateral trade between China and Russia.

2. The geographical proximity of China and Russia is central to increasing the volume of bilateral trade. The coefficient of the geographic distance between the independent variables is −0.601511, this negative number is consistent with the expected sign, indicating that every 1% increase in the distance the between two countries will decrease the logarithm of the volume of bilateral trade between these countries by 0.601511%. It can be seen that the distance between the two countries is a critical factor limiting the economic development of the two countries. The farther the distance, the higher the transportation cost will be, and the profit of both sides will be smaller; therefore, China should actively strengthen economic and trade cooperation with neighboring countries, thereby reducing transportation costs, strengthening trade cooperation between Northeast China and the Russian Far East and promote the economic development of the two countries [14]. Although Chinese and Russian capitals are far apart, the currently discussed Sino-Russian Free Trade Zone is located in Heilongjiang Province, the northernmost province of China, closest to the Russian Far East, which minimizes the transportation costs between China and Russia when doing trade. Thus, China and Russia should actively take advantage of this location and the unique convenience that China and Russia are the largest neighboring countries, so that the free trade area between China and Russia is significant for the two countries.

3. The signing of a free trade agreement between China and Russia will promote the increase in the volume of bilateral trade (0.177371). The coefficient of the independent variable showing whether two countries have signed a free trade agreement is 0.177371, which indicates that if this value is 1, that is, two countries have signed a free trade agreement; the volume of bilateral trade of the two countries will increase by 0.177371%. There may be two reasons why the effect of the model is not particularly significant: first, the sample size selected for modeling is limited, and there are relatively few countries that have signed a free trade agreement with China; secondly, the time for China to sign free trade agreements with other countries is relatively short, and some free trade zones have just been established; therefore, their impact on bilateral trade is negligible. Still, it can be seen that the creation of a free trade zone between China and Russia will increase the volume of bilateral trade between the participants. Thus, it is undeniable that the signing of a free trade agreement actually contributes to the growth of bilateral trade. The creation of a free trade zone between China and Russia will have a positive effect on the Sino-Russian trade.

4. The Belt and Road Initiative increased the volume of bilateral trade (0.112503). The coefficient indicating whether the BRI is proposed equals 0.112503, which indicates that if $jB$ is 1, that is, the proposed BRI will benefit the economic development of countries along the route, and the volume of bilateral trade between the two countries will increase by 0.112503. Since the Belt and Road Policy was proposed only in 2013, the time from its proposal is relatively short, and there will be a backlog in policy implementation and project launch; therefore, the level of significance is not obvious. It can still be argued that the Belt and Road Initiative has contributed to the economic development of countries.

5. The coefficient of foreign direct investment in China for the explanatory variable (0.003874) indicates that for every 1% increase in direct investment in China by a trad-

ing partner, the logarithm of bilateral trade between the two countries will increase by 0.003874%. It can be seen that foreign direct investment positively influences the volume of bilateral trade, but not as much as GDP growth, and has the weakest effect among all indicators; therefore, in the process of trade and investment cooperation between the two countries, attention should be paid to increasing the breadth and depth of investment between trading partners [14,141–143].

6.  In 2023, due to the ongoing COVID-19 epidemic, China experienced a new wave of the pandemic. The Chinese authorities are forced to maintain a high level of restrictions, including those related to business contacts. Sooner or later, the epidemic will cease to be a deterrent, but it prevents the immediate development of cooperation that requires intensive human contacts.

Chinese business fears secondary sanctions, as well as administrative and criminal prosecution by the US authorities in case of violation of the US sanctions regime, and restrictive measures of other countries. This situation may arise, for example, in the case of mutual settlements between Chinese companies and Russian counterparties under sanctions in US dollars or even euros. Another scenario is the supply to Russia of goods that are produced in China under an American license and at the same time falls under US export control (for example, electronics). The resonant criminal and administrative prosecution by the US authorities of the Chinese company ZTE apparently had a serious psychological impact on the Chinese business. The United States accused ZTE of supplying equipment with American components to Iran without permission and bypassing the export control regime. As a result, ZTE pledged to pay more than $1 billion US dollars in fines to several US government agencies. The U.S. authorities' attempt to prosecute Huawei CFO Meng Wanzhou had a similar effect [144].

We can talk about the same effect in connection with the blocking sanctions of the US Treasury against the Chinese company COSCO SHIPPING Tanker for the alleged transportation of Iranian oil (however, the company could quickly get out of the sanctions administratively). The risks of secondary sanctions and coercive measures are forcing Chinese businesses to carefully evaluate options for cooperation with Russia. A particularly thorough analysis is carried out by companies that are actively working in the US and EU markets.

Simultaneously, secondary sanctions and coercive measures alone are unlikely to stop the growth of trade relations between Russia and China in the new conditions. Export control of foreign countries does not apply to those goods that China produces using its own technologies. And there are more and more commodities like this. Financial sanctions are unlikely to affect Russian and Chinese businesses in the event of transactions in the yuan outside the contours of the American financial system. That is, trading in national currencies will mitigate their impact. The Chinese authorities are actively modernizing their legislation aimed at protecting Chinese firms from Western sanctions. Undoubtedly, the risk of secondary sanctions and enforcement measures will be significant in the medium term. The Russian business should be sympathetic to the caution of Chinese partners. However, operational work on financial mechanisms for mutual settlements and the development of market niches not related to Western technologies will provide more opportunities in the long term.

The knowledge of the Chinese language, culture, and law is an important fundamental factor for further cooperation. The lack of such competencies will prevent Russian business from looking for markets in China, attracting Chinese investments and suppliers, and conducting effective negotiations. Chinese businesspersons in Russia, for their part, are quickly mastering the Russian language. The development of cultural competencies, at first glance, is secondary compared to financial infrastructure, transport corridors, and other conditions [144].

## 5. Discussion

The results can be divided into two themes: the first one discusses the sustainable development of the bilateral trade association between China and Russia. The results of the secondary analysis in the context of foreign policies and interests showed that both Russia and China sustainably develop economic relations due to similarities in policies, reforms, and political systems. This complies with the findings of recently published studies such as Chen and Bao [35] and Mahlstein et al. [36] showcasing that both Russia and China expand their bilateral ties, and considerably impact sustainable economic growth. Additionally, the economic ties between the two nations are strong having a trading volume of approximately USD150 billion, implying that sustainable economic development has a positive impact on the global economy. However, the findings of the current study are novel regarding the power structure and foreign policies of the two countries, which helps them have strong bilateral ties at the international level, promoting peace, security, economic trade, and development.

Besides, the first theme also focused on economic ties and bilateral trade relations showing strong ties between China and Russia due to mutual reassurances and ease of trade barriers. Both countries have developed bilateral ties in various areas such as defense, aerospace, technology, energy, and goods. Moreover, after the Ukraine crisis in early 2022, China eased trade restrictions on Russian wheat, as Russia became a major wheat supplier and to cover up the wheat shortage in China, this move was important and helpful, showing that the political crisis can help further increase the bilateral connection between the two countries. It is projected that both countries have plans to enhance their trade volume in 2024-2025 considering the dependence on each other in terms of economic development and trade of essential items. This finding contradicts what has been found in the recent literature. For instance, in the context of the Ukrainian crisis, Mahlstein et al. [36] found that non-associated economies and countries received significant benefits during the crisis; however, China was in support of the embargo against Russia, which led to the loss of the Russian economy.

The second theme of the findings focused on the impact of sustainable development of the Sino-Russian trade relationship on Europe and the US under international sanctions. In relation to economic sanctions against Russia and its impact on the US and Europe, the findings showed that the sanctions disrupted Russia's real economy – the production, sale, and transportation of goods. However, the US faced little or no impact from the sanctions; however, the sanctions did have negative implications for the US companies operating in Russia. For instance, the energy crisis in the USA worsened due to gasoline prices. Similarly, the EU become its major trade partner of Russia and particularly was reluctant to impose sanctions on Russia. However, due to sanctions, the US was pushed to strengthen its ties with China. In line with this, the study by Liadze et al. [65] agrees that international sanctions placed on Russia negatively impact the region and has caused a considerable impact on the rest of the world along with the Russian economy. The exchange rate was also affected by the crisis, and EU countries encountered an increase in the consumer price index because of commodity shortages, particularly energy prices elevated noticeably, which enhanced the cost of production in several industries, and the impact was to be borne by consumers [76].

Regarding the impact of the Sino-Russian relationship on the US, the findings showed that association and coordination between China and Russia undermine the existing international order, bringing several challenges with reference to the strategic areas of the US. Relating to the increasing participation and coordination between the two countries is viewed to be a major challenge for the US, and policymakers and residents of the US also believe that the US is likely to lose its dominance and power in various regions of the world. Additionally, Americans view the increasing corporation of China with leading countries as a threat to the power and rule of America globally. On the other hand, the policies of EU countries do not support or favor Russia, which provides a competitive edge to China in

comparison to the US. Therefore, if processes continue at the same pace, the US can have long-lasting consequences.

In the context of the Sino-Russian relation impact on EU countries, the findings showed that the increase in the Sino-Russian association has a negative impact on European industries, and most of the industries because of the relationship between these two countries comprises electronic machinery, nuclear reactors, machinery, and equipment. The findings further showed that the Sino-Russian relationship and its impact on the EU can be further categorized into short-term and long-term impacts. For instance, it will challenge for the EU countries to increase exports as China is the major competitor and has an advantage over it. However, eventually, it can bring opportunities for EU countries, and their close relationship and ties will be useful for Europe in the expansion of trade and entering new markets. Therefore, the Sino-Russian relationship in the short run is adversely affecting EU countries in relation to the reduction of their exports.

Previous literature in the context of the Sino-Russian relation impact on the US strongly correlates with the current findings, agreeing that the US considers both China and Russia as rivals, and that the strong and bilateral ties between them can have a significant impact on the western countries along with the US. Megits [31], in this context, found that the economic affiliation between Russia and China are capably adversely affecting the national interest of the US in the region and seriously dent the US hegemony. Overall, there is limited literature that has focused on the Sino-Russian impact on the US and European countries; however, this study has contributed to explaining the impact of this relationship, which can be used as evidence for further examining the relationship from other economic and trade parameters. The hypothesis of the study, namely, "There is a significant and adverse influence of China-Russia trade relations development on the European countries", remains accepted based on the findings of the study.

International labor specialization is deepening in Russia-China relations, asymmetric interdependence is increasing, and it is overcoming possible through development, diversification and improvement of institutional forms, methods, and directions of financial and economic cooperation.

In terms of imports, Russia has become one of the most China-dependent economies. For Chinese importers of raw materials and exporters of goods, however, the situation is profitable. Imports from China are insufficient to make up for the deficit of Western components. Supplies to Russia of Chinese goods that are on the EU or US restrictive lists are growing more slowly than products free of sanctions (e.g., consumer electronics). Over time, Chinese companies will be able to replace some commodities that Russia cannot buy from the West now. But this will take time and will face difficulties. As for the most technologically advanced products, China still cannot produce microchips at the same level as Taiwan or South Korea. Thus, Russia-China trade relations will assist Russia in overcoming the influence of sanctions. Russia-China trade is likely to positively influence Russia's economics.

One thing is evident that international sanctions will influence a country's GDP. The IMF and the World Bank assert that the country will experience a decline in imports and exports in 2023 due to sanctions [111]. Moreover, organizations and firms' performance are expected to decline due to increase in raw material and energy cost and low demand influenced by the international sanctions. Inflation in Russia and around the globe is likely to increase due to sanctions that stop Russia from supplying fuel and energy.

The areas of strategic cooperation between China and Russia will largely determine Russia's power and prosperity. The growing development of Siberia and the Far East will improve Russian exports and will supply budget sustainability and economic stability, despite the pressure of sanctions imposed by the West. It will reorient the Russian oil supplies. The critical success factor is the development of the Soyuz Vostok gas pipeline that will go through East Siberian regions and end at production areas on Yamal with China.

The following measures are proposed as promising forms of diversification of financial and economic cooperation between Russia and China: the development of public-

private partnerships, the creation of joint ventures in the two countries, the implementation of the "One Belt, One Road" project, intensifying financial cooperation within the NDB and AIIB, diversifying mutual trade and industrial cooperation within the energy sector toward the improvement of raw materials production technologies and development of nuclear energy, and increasing the share of industrial products. Thus, "One Belt, One Road" project is likely to positively influence Russia and China economic cooperation [14,142].

One of the key findings of the study is that China and Russia plan to increase the volume of bilateral trade in the next two years. The findings illustrate the similarities between the two countries in terms of political systems, reforms, and policies. Both countries have centralized power structures that help them support each other in international forums, such as the UNSC. However, while Russia works to promote security and political stability, China focuses more on trade and economic development. These differences could present some challenges to the development of the Sino-Russian economic relationship, but they are not insurmountable.

This article also examines the impact of the Sino-Russian economic relationship on the US and Europe under international sanctions. The findings indicate that US companies have been negatively impacted due to sanctions on Russia, while European countries-particularly Germany-have been reluctant to take steps against Russia fearing economic implications. As a result, the economic ties between Russia and China have increased. It is disclosed that the close coordination and association between China and Russia undermine the US-led international order, increasing challenges for the US, and even threatening the rule and power of the US due to increased cooperation between China and Russia. Similarly, regarding the impact on European countries, the findings show that the cooperation between China and Russia has a negative impact on European industries such as machinery, nuclear reactors, and electronic machinery.

Furthermore, the study revealed that EU countries tend to be complementary to Russia in the Chinese market. In other words, the reduction in bilateral import tariffs may most probably reduce the EU exports to China and Russia significantly. Therefore, the development of this relationship will adversely impact the European Zone. However, eventually, the European countries can explore other markets and improve their economic conditions.

This study highlights the acute relevance and significance of the topic, which is determined by the ongoing major geopolitical crisis in the international relations of Russia and some countries of the world. This crisis inevitably affects regional processes in the Eurasian region. Under the escalating confrontation of sanctions, China will increase its presence in the Indo-Pacific region during the redistribution of influence in the context of multiple sanctions against Russia. Further growth of trade and economic cooperation between countries, both on a bilateral and multilateral basis, contributes to a more complete unlocking of the potential of national economies, increasing their competitiveness and helping actively interacting states to enter a phase of advanced and sustainable development.

The findings highlight the importance of the Sino-Russian trade relationship for both countries and its potential impact on the global economic order. This study highlights the importance of the Sino-Russian trade relationship for both countries. China and Russia have been able to leverage their similarities in political systems, reforms, and policies to build a strong economic relationship that benefits both parties. This study suggests that both countries plan to increase their trade volume in the next two years, indicating their commitment to further strengthening their economic ties. Besides, the findings point out the impact of the Sino-Russian trade relationship on the US and European countries. The US has been affected by the sanctions it has imposed on Russia, as US companies have lost business opportunities. On the other hand, European countries have been reluctant to take steps against Russia due to their strong economic ties, particularly with Germany. As a result, the economic ties between Russia and China have strengthened further. It is also illustrated that the negative impact of the Sino-Russian trade relationship on certain industries in Europe, such as machinery, nuclear reactors, and electronic machinery. This

study suggests that the reduction in bilateral import tariffs may significantly reduce EU exports to China and Russia, which would have adverse effects on the European Zone.

The study makes a significant contribution to the understanding of the future of the Sino-Russian economic relationship. The report's findings provide empirical evidence of the potential opportunities and challenges facing the development of this relationship under the ongoing global turbulence, the post-COVID situation, and sanctions pressure. The report's theoretical and practical contributions provide important insights into the economic and geopolitical implications of this relationship in the US, Europe, and other regions of the world. This study's findings will be useful for policymakers, scholars, and practitioners interested in the future of the Sino-Russian economic relationship.

## 6. Conclusions and Policy Recommendations

### 6.1. Findings of the Study

The current study has multifaceted novelty, significance, and theoretical and practical contributions. The findings in relation to the sustainable development of the economic relationship between Russia and China showed similarities between the countries in terms of political systems, reforms, and policies. The centralized power structures present in the two countries are found to overlap, thus helping support each other in international forums like UNSC. Nevertheless, China focuses more on trade and economic development and trade, whereas Russia works to promote security and political stability. In terms of bilateral trade and economic ties, it is found that both countries plan to increase trade volume in 2024–2025.

Relating to the impact of the development of the Sino-Russian trade relationship on the US and Europe under international sanctions highlighting that US companies were affected due to sanctions on Russia, while European countries being one of the major trade partners with Russia, especially Germany, were reluctant to take steps against Russia fearing economic implications. Hence, the economic ties between Russia and China increased as a consequence of this. In terms of the Sino-Russian relationship impact on the US, it was found that the close coordination and association between China and Russia undermine the US-led international order, increasing challenges for the US, and even threatening the rule and power of the US due to increased cooperation between China and Russia. Similarly, regarding the impact on European countries, the findings showed that the cooperation between China and Russia had a negative impact on European industries such as machinery, nuclear reactors, and electronic machinery, etc. In terms of short-term impact, European countries will have consequences; however, eventually, they can explore other markets and improve economic conditions.

Overall, it is observed that EU countries tend to be complementary to Russia in the Chinese market. In other words, it is found that the reduction of bilateral import tariffs may most probably reduce the EU exports to China and Russia very significantly. Therefore, the development of this relationship will adversely impact the European Zone.

### 6.2. Strengths and Limitations

A qualitative approach was applied using secondary sources for data collection, such as government sources, journal articles, and websites. A thematic analysis was applied for data analysis, and it was found that Russia and China centralized the power and sustainable development of economic ties between them. China focuses more on economic development and trade, while Russia emphasizes more the security and promotion of political stability. The cooperation between China and Russia negatively affects the power and influence of the US, while the impact of sanctions on Russia has negative implications for US companies and European countries in the short term. However, the long-term ties between Russia and China can bring opportunities for European countries to explore other markets. The study has limitations in terms of findings that are mainly in the general context. Future research can consider specific sectors in the US and Europe that affected the Sino-Russian trade relations and the impact of anti-Russian sanctions.

The key study strength includes the successful accomplishment of the research objectives, which were not addressed comprehensively in previous literature. The use of a range of secondary sources has provided a broader overview of the US and China's bilateral economic and trade relations and their impact on the US and European countries. Nevertheless, the study has limitations in terms of scope and generalizability because specific factors and sectors affected by the Russian-Chinese cooperation are not covered in this stud; therefore, the findings were more in a general context rather than a specific impact on the US and Europe.

An attempt has been made to conduct a detailed and in-depth analysis of the impact of Russian-Chinese trade, but there are still some shortcomings, mainly: first, data collection for each indicator is not particularly homogeneous, and data are collected from different websites. There are discrepancies in the data, which will lead to some errors in the results of the calculations. Second, in terms of research methods, the trade gravity model is more inclined to examine factors affecting bilateral trade potential and cannot measure specific trade effects in detail.

### 6.3. Recommendations for Future Research

*Theoretical significance:* based on relevant data from the UN commodity trade database and the World Bank database, the paper conducts an inductive analysis of China's and Russia's current economic development situations. In response to the above analysis and summary of problems in the development process of Russia and China, the structure of Russia's exports to China is unified; the structure of Russia-China commodity trade is fragile, Chinese goods face stiff competition in the Russian market, and the potential of complementary trade between Russia and China is not fully exploited. To enrich the empirical results of the theoretical research on the competitiveness and complementarity of Sino-Russian trade, we hope to provide some possible suggestions for developing trade between Russia and China and to promote the rational development of the trade structure. Based on the data from 2008 to 2022, the gravity model is used to empirically analyze the impact of Russia-China trade to measure the main factors affecting the trade flow between Russia and China.

*Practical Importance:* With the ongoing economic sanctions of Western countries, Russia and China should increase cooperation. This article analyzes the competitiveness and complementarity of the import-export structure of commodity trade between Russia and China. Based on the results of the empirical analysis of the trade gravity model, relevant proposals are put forward to help solve the problem of unbalanced development of Russia-China trade, improve the level of its development, significantly improve and optimize trade structures between Russia and China.

Based on these findings, it is proposed to consider the factors such as manufacturing, oil and gas, electronics, and automobiles and services sectors of the US and European countries such as the UK and Germany to evaluate the impact of Russia and China trade and bilateral cooperation and specifically the impact of sanctions against Russia on European countries. Emphasizing these factors in research will help generalize the research findings using more authentic and primary sources such as government databases and financial statements published by trade ministries of the US and European countries to evaluate the impact of anti-Russian sanctions and Sino-Russian cooperation on various sectors.

### 6.4. Policy Recommendations

The sustainable development of the economic relationship between Russia and China requires a comprehensive and coordinated approach, taking into account their respective strengths, priorities, and challenges, as well as the potential impacts on other countries and the international order. The policy recommendations proposed in this study aim to provide a conceptual framework and practical guidance for policymakers, business leaders, and civil society actors in both countries and beyond:

- Promote trade diversification: While Russia and China are increasing their bilateral trade volume, they should also diversify their trade partners and export markets to reduce their dependence on each other. This will not only reduce the risk of exogenous shocks to their trade relationships but also enhance their competitiveness in the global market.
- Enhance financial cooperation: To further reduce their dependence on the US-dominated financial system, Russia and China should strengthen their financial cooperation, including increasing the use of their own currencies in bilateral trade, expanding the scope of their swap agreements, and promoting the use of digital currencies in cross-border transactions.
- Address trade barriers: To increase trade volume and reduce trade costs, Russia and China should address existing trade barriers, such as non-tariff measures and technical standards, and negotiate a free trade agreement. This will help unlock the potential of their national economies and enhance their competitiveness in the global market.
- Strengthen political and security coordination: As two major powers with different strengths and priorities, Russia and China should strengthen their political and security coordination to address common challenges, such as terrorism, regional conflicts, and cybersecurity. This will help promote mutual trust and understanding, and enhance their collective bargaining power in the international arena.
- Mitigate negative impacts on other countries: While the cooperation between Russia and China may have a negative impact on other countries, such as the US and some European countries, the two countries should take measures to mitigate these impacts, such as providing compensation or alternative markets. Moreover, they should communicate with other countries in a transparent and constructive manner to avoid misunderstandings and conflicts.
- Promote sustainable development: To ensure the long-term sustainability of their economic relationship, Russia and China should prioritize sustainable development, including environmental protection, social responsibility, and green finance. This will not only enhance their international image and reputation but also contribute to the common goals of the international community.
- Foster people-to-people exchanges: To enhance mutual understanding and trust between their peoples, Russia and China should promote people-to-people exchanges, including cultural, educational, and tourism activities. This will help build a solid foundation for their long-term cooperation and friendship.

*6.5. Managerial Implications*

Based on the findings of this study, there are several managerial implications that can be drawn for companies operating in Russia, China, and other countries impacted by the Sino-Russian trade relationship:

- Diversification of markets: Given the current geopolitical situation, companies should consider diversifying their export markets to reduce their reliance on any single market. This would help mitigate the risks associated with changes in trade relations between countries, such as the ongoing tensions between Russia and the West.
- Strategic partnerships: Companies should consider forming strategic partnerships with local companies in China and Russia to enhance their understanding of the local market and regulatory environment. This can help companies navigate the complexities of operating in these markets and build relationships with key stakeholders.
- Adaptation to changing regulations: Companies should closely monitor changes in trade regulations and adapt their operations accordingly. The findings of this study suggest that trade relations between countries are subject to ongoing changes due to the complex geopolitical environment. Thus, companies should be prepared to quickly adapt to new regulatory regimes to avoid disruptions in their operations.

- Innovation and technology adoption: Companies should continue to invest in innovation and technology adoption to improve their competitiveness in the global market. This is particularly important given the findings of this study, which suggest that China is increasingly focusing on economic development and trade and is likely to continue to grow as a global economic power.
- Risk management: Finally, companies should have robust risk management policies and procedures in place to mitigate the risks associated with geopolitical tensions and fluctuations in trade relations. This includes developing contingency plans for potential disruptions in supply chains and taking steps to ensure financial stability in the event of economic shocks.

**Author Contributions:** Conceptualization, G.A., R.L., Q.A., H.F. and M.V.; Methodology, Q.A., S.P., I.E. and M.V.; software, G.A., validation, R.L. and Q.A.; formal analysis, Q.A.; H.F., I.E.; investigation, V.P., M.K. and I.E.; resources, Q.A. and H.F.; data curation, G.A., R.L. and Q.A.; writing—original draft preparation, G.A., R.L., Q.A., H.F., V.P., I.E., M.K., S.P. and M.V.; writing—review and editing, G.A., Q.A. and M.V.; visualization, M.K. and G.A.; supervision, G.A., Q.A. and M.V.; project administration, G.A., Q.A. and M.V.; funding acquisition, not applicable. All authors have read and agreed to the published version of the manuscript.

**Funding:** This research received no external funding.

**Institutional Review Board Statement:** Not applicable.

**Informed Consent Statement:** Not applicable.

**Data Availability Statement:** Not applicable.

**Conflicts of Interest:** The authors declare no conflict of interest.

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
