# Peer review of "Development of Trade and Financial-Economical Relationships between China and Russia: A Study Based on the Trade Gravity Model"

_sustainability, doi:10.3390/su15076099_

Round 1

Reviewer 1 Report

The manuscript titled “Development of Trade and Financial-Economical Relationships between China and Russia: A Study Based on the Trade Gravity Model” is reviewed. My comments are reported below;

1.     In Figures X and Y axes are not clear to the readers. The authors might use different colors for these axes.

2.     The authors should highlight and describe the motivation of the study, explain the gaps in the literature, and the importance of the study to the literature. This is also partially missing in the manuscript.

3.     The authors have 11 pages of introduction and literature, also 38 pages single spaces manuscript, the authors should little bit minimize the size of the manuscript, especially the introduction and literature review sections.

4.     Within 38 pages, discussion and empirical findings hold only 3-4 pages which is not enough.  

Reviewer 2 Report

Comment 1: The authors should clarify the meaning of Figure 1-4 . Is it importance and relevant? 

Comment 2: The study uses the trade gravity model to study the financial-economical relationships between China and Russia. I  think this research question is not new. The authors should highlight the difference from the previous study. Whether there is exogenous shock? This is necessary. 

Comment 3: The literature review is poor, What's the contribution of this paper? 

Comment 4: The authors should highlight the main research questions. There are too many tables and figures. It's difficult to understand the intuition.

Comment 5: The managerial insights are poor.

Comment 6:The writing should be improved. 

Comment 7: Where is the data source? Is it avaiable and public?  

Comment 8:The figure is not professional. Please simplify those figures.

Reviewer 3 Report

"Development of Trade and Financial-Economical Relationships between China and Russia: A Study Based on the Trade Gravity Model" is a nicely presented and well-explained article. I suggest the following improvements:

1. Please refer to the "Instructions for Authors" of the journal available at: https://www.mdpi.com/journal/sustainability/instructions and rewrite the abstract as per the instructions below:

  • Abstract: The abstract should be a total of about 200 words maximum. The abstract should be a single paragraph and should follow the style of structured abstracts, but without headings: 1) Background: Place the question addressed in a broad context and highlight the purpose of the study; 2) Methods: Describe briefly the main methods or treatments applied. Include any relevant preregistration numbers, and species and strains of any animals used. 3) Results: Summarize the article's main findings; and 4) Conclusion: Indicate the main conclusions or interpretations. The abstract should be an objective representation of the article: it must not contain results which are not presented and substantiated in the main text and should not exaggerate the main conclusions.

2. Line 65: Please avoid writing a single sentence. Line 76-79 Paragraph is too short.

3. Introduction section is too short and lacks what specific research question this study intends to investigate.

4. Please mention units in the Figures.

5. Section 2, Theoretical Framework represents a detailed review of the literature. please be specific about the theory and how the theory is employed in the current study. 

Reviewer 4 Report

Peer Review for "Sustainability"

Manuscript Title: Development of Trade and Financial-Economical Relationships between China and Russia: A Study Based on the Trade Gravity Model

OVERVIEW:

This research is relevant and focuses on the present affairs in our political and economic sphere. Specifically, it is based on the trade gravity model and was performed with regard to the latest trends in

research, coupled with the current challenges and trends in the global economy that we are experiencing in today's time.

              Methods of grouping and generalization, as well as scientific induction and deduction are performed.  Comparative, systemic and statistical analysis, as well as econometric modeling and forecasting, are executed in order to arrive to its conclusions. The thesis of this study is to use the gravity model of international trade to examine linkages between China and Russia.

SUGGESTIONS FOR IMPROVEMENTS:

[1] I strongly recommend the authors re-write the abstract to make it more succinct, focused, and to highlight precisely what the study seeks to accomplish and what the findings are.  As it reads now, it is very verbose and not specific.  A reader should be able to read the abstract and understand what the paper is about and what its contribution is.  This current abstract, in my view, does not achieve this purpose.

[2] In the Data section of the paper, discuss more the sources for the data.  This will be helpful to readers.

[3] All figures should be more precise (and not use, say, Excel, to create the graphics).

[4] Overall, the manuscript will require professional proof-reading.

[5] Discuss more the RCA equation (in Equation 9) and how it can help model interdependencies between the two countries.

Round 2

Reviewer 2 Report

Dear Editors, 

I still have some comments.

Comment 1: The language should be improved.

Comment 2: The economic insights should be explained.

Comment 3: The literature review is not good.

Comment 4: What's the poliy implications of this paper? The authors should summarize those findings in abstract and conclusion part. 

Thanks

Reviewer 4 Report

This manuscript is very interesting and makes a nice contribution to the literature.  The revisions are appropriate and well-done for this paper.  I commend the author/s for their work and for these revisions.

Author Response

Dear Reviewer,

Thank you very much for your positive feedback.

Sincerely yours,

Qamar Abbas

Round 3

Reviewer 2 Report

Dear editors, 

The authors have addressed my all concern.  Some recent literatures should be cited. 

Li C., Liu Q., Zhou P. , & Huang H. (2021). Optimal innovation investment: The role of subsidy schemes and supply chain channel power structure. Computers & Industrial Engineering, 157, 107291.

Zhai Y., Bu C., & Zhou P. (2022). Effects of channel power structures on pricing and service provision decisions in a supply chain: A perspective of demand disruptions. Computers & Industrial Engineering, 173, 108715.

Chen J., Sun C., Wang Y., Liu J., & Zhou P. (2023). Carbon emission reduction policy with privatization in an oligopoly model. Environmental Science and Pollution Research, 1-22.

https://doi.org/10.1007/s11356-022-24256-2

Thanks
